# E-cadherin endocytosis promotes non-canonical EGFR:STAT signalling to induce cell death and inhibit heterochromatinisation

Miguel Ramirez Moreno[1¤a], Amy Quinton[1], Eleanor Jacobsen[1],
Przemyslaw A. Stempor[2], Martin P. Zeidler[1], Natalia A. Bulgakova[1¤b]*

**1** School of Biosciences, University of Sheffield, Sheffield, United Kingdom, **2** Nonacus Limited, Birmingham, United Kingdom

¤a School of Biological Sciences, University of Southampton, Southampton, United Kingdom
¤b School of Biological and Behavioural Sciences, Queen Mary University of London, London, United Kingdom
* n.bulgakova@qmul.ac.uk

## Abstract

Signalling molecules often contribute to several downstream pathways that produce distinct transcriptional outputs and cellular phenotypes. One of the major unanswered questions in cell biology is how multiple activities of signalling molecules are coordinated in space and time in vivo. Here, we focus on the Signal Transducer and Activator of Transcription (STAT) protein as a paradigm of signalling molecules involved in several independent signalling pathways. Using *Drosophila* wing discs as an in vivo model, we demonstrate an interplay of at least three STAT activities in this tissue. In addition to the 'canonical' pathways, in which STAT is phosphorylated and activated by Janus Kinases, STAT is involved in two 'non-canonical' pathways. In one pathway, STAT is activated by the Epidermal Growth Factor Receptor (EGFR), promoting apoptosis. In the other, it binds the Heterochromatin Protein 1 (HP1) to enhance heterochromatin formation. We provide evidence that while the 'canonical' STAT signalling is dominant over 'non-canonical' pathways, EGFR:STAT and HP1:STAT pathways compete for the availability of unphosphorylated STAT. We also describe a central role for the cell-cell adhesion protein E-cadherin, with both EGFR and STAT colocalising with E-cadherin at cell-cell junctions and on intracellular vesicles. We show that elevated intracellular E-cadherin promotes EGFR:STAT pathway leading to apoptosis, which is prevented by inhibiting E-cad endocytosis. Taken together, we conclude that E-cadherin controls the balance between two non-canonical STAT activities. We hypothesise that this balance represents a tumour-suppressive mechanism, in which junctional disassembly in dysregulated epithelial-to-mesenchymal transitions would shift this balance towards the EGFR:STAT signalling to promote apoptosis.

**Data availability statement:** The RNA sequencing data is available at PRJNA1185032 https://www.ncbi.nlm.nih.gov/bioproject/1185032. All microscopy data is available at the Queen Mary Research Online repository, https://qmro.qmul.ac.uk/xmlui/handle/123456789/101582

**Funding:** This work was supported by grants from the Biotechnology and Biological Sciences Research Council, UK Research and Innovation BB/P007503/1 (NAB) and BB/W019698/1 (School of Biological and Behavioural Sciences, Queen Mary University of London). MRM received salary support from BB/P007503/1. The funders had no role in study design, data collection and analysis, decision to publish, or preparation of the manuscript.

**Competing interests:** The authors have declared that no competing interests exist.

## Author summary

One of the central questions of modern biology is how a small number of highly conserved signal transduction pathways act and interact to generate the cellular and transcriptional diversity required for complex multicellular life. Here, we utilise the low complexity *Drosophila* model organism to study the Signal Transducer and Activator of Transcription (STAT) protein signalling in a developing in vivo tissue using developmental genetics approaches. We demonstrate the existence of a hierarchy of at least three STAT-related signalling pathways, with one pathway dominating the other two, which, in turn, compete for STAT availability. Additionally, we show that this competition is controlled by the protein E-cadherin, which is crucial for adhesion between cells in epithelial tissues and for maintaining tissue integrity. In particular, E-cadherin internalisation from the cell surface into the cell enhances one of the STAT downstream pathways that leads to cell death. Consequently, experimentally blocking E-cadherin internalisation prevents this cell death. We hypothesise that balancing the competition between two STAT downstream pathways through E-cadherin serves a tumour-suppressive role; specifically, the disassembly of adhesion by cells undergoing cancerous transformation shifts this balance towards the pathway that induces cell death.

## Introduction

Signalling pathways control numerous aspects of cellular behaviour, including proliferation, differentiation, morphological changes and cell death. Many of these molecular pathways transduce signals from the environment into the cell nucleus, leading to changes in gene expression and chromatin organisation. However, pathways do not have single fixed outcomes but produce a range of qualitatively different transcriptional outputs depending on the temporal, spatial and environmental context of the cell [1]. One of the major unanswered questions in cell biology is how context-appropriate signalling outputs are selected and coordinated by cells [1].

The Janus Kinase/Signal Transducer and Activator of Transcription (JAK/STAT) pathway delivers its effects through several parallel modalities. In 'canonical' JAK/STAT signalling, ligand stimulation of cytokine receptors causes the activation of receptor-associated JAKs, leading to the phosphorylation of latent STAT transcription factors on highly conserved tyrosine residues. This induces STAT dimerisation and translocation to the nucleus, where the complex directly binds to DNA and activates transcription [2,3]. However, STAT has at least two additional non-canonical activities. In the first of these, non-canonical STAT modulates chromatin organisation [4]. In this pathway, STAT not phosphorylated on the canonical C-terminal tyrosine residue binds to Heterochromatin Protein 1 (HP1) and stabilises heterochromatin, a function conserved in both *Drosophila* and mammalian cells [4–7]. While the binding between STAT and HP1 has a tumour-suppressor role [6,7], the cellular function of heterochromatin stabilisation by STAT:HP1 interaction is unclear. In the second non-canonical

STAT activity, the direct activation of STAT1, STAT3 and STAT5 by the Epidermal Growth Factor Receptor (EGFR) promotes apoptosis in human cell lines [8–10]. In this pathway, EGFR phosphorylates STAT5 at unique sites distinct from those of the canonical pathway [11].

There is growing evidence that cell-cell adhesion within epithelia, as mediated by molecules such as the transmembrane protein E-cadherin (E-cad) [12], contributes to the diversity of transcriptional outputs by modulating various signalling events, thereby influencing cell proliferation, differentiation and apoptosis [13–16]. For example, E-cad binds the receptor tyrosine kinase EGFR in both mammalian and fly cells, promoting both inhibition and activation of canonical EGFR signalling in a context-dependent manner [14,16,17]. E-cad has also been linked to STAT protein localisation and canonical signalling in *Drosophila,* with both STAT and E-cad co-immunoprecipitating and binding Par-3 [18–20]. Finally, E-cad overexpression is sufficient to suppress position effect variegation in the developing *Drosophila* eye [21], suggesting that E-cad may also be able to affect heterochromatinisation in vivo.

Extensive evidence highlights that endocytosis can also change signalling pathway activities and outputs [22–25]. The internalisation of receptors from the plasma membrane into early endosomes creates intracellular signalling platforms, or "signalosomes", with high concentrations of pathway components to facilitate signal transduction. As such, endocytosis leads to the compartmentalisation of signalling activities within physically distinct domains inside cells, enabling the specificity and robustness of the signalling response to be controlled [22]. Indeed, it is well established that both the canonical JAK/STAT and EGFR signalling pathways are affected by endocytosis, with recent evidence indicating that endocytosis acts to both prolong signalling activity and quantitatively alter signalling outputs [24,26,27]. Thus, endocytosis is essential for the expression of some, but not all, JAK/STAT pathway target genes. Similarly, endocytic trafficking plays a major role in the regulation of EGFR signalling activities [16,28,29]. Endocytosis can both enhance and inhibit EGFR signalling depending on the ligand concentration and specific endocytic pathway involved [28,29]. Relevant to this work, non-canonical STAT3 signalling downstream of EGFR is also regulated by endocytosis, with the PYK2 kinase being recruited to sustain this signalling on early endosomes [30].

The regulation of signalling by cell-cell adhesion and endocytosis cannot be considered in isolation. In both mammalian and fly cells, E-cad is constantly endocytosed and then recycled in Rab11-positive recycling endosomes [31,32]. The p120-catenin (p120ctn) protein is the key inhibitor of E-cad endocytosis in both mammalian and fly cells [31,33–35], and its overexpression is sufficient to prevent E-cad from being internalised [34,35]. Given this endocytic turnover, E-cad trafficking may directly affect signalling pathways. For example, E-cad and Par-3 co-endocytose together in the *Drosophila* embryonic epidermis [19], which opens the possibility that other interacting molecules – including STAT and EGFR – are also being co-internalised. We therefore hypothesised that cross-talk between cell-cell adhesion and endocytic processes potentially provides an additional level of signalling regulation, which can be investigated by forcing the intracellular accumulation of E-cad. In *Drosophila* wing imaginal discs, E-cad intracellular accumulation due to enhanced endocytosis induced by knocking down the Clathrin adaptor protein AP-1 is sufficient to increase the activity of c-Jun NH2-terminal kinase (JNK) signalling and cellular apoptosis [36]. This increase in apoptosis during larval stages leads to a 'small wing' phenotype in adults – a phenotype that was rescued by inhibiting E-cad endocytosis using p120ctn overexpression [36].

In this report, we use *Drosophila melanogaster* – a model organism that has been central to dissecting the developmental genetics principles of key signalling pathways [37] – to genetically elucidate the in vivo functions and regulation of non-canonical STAT signalling. This is facilitated by the low complexity of the fly genome, which allows us to manipulate each pathway without redundancies. Here we demonstrate that two non-canonical STAT activities – mediated via interactions with EGFR and HP1 – act concurrently and compete for STAT availability. We show that STAT and EGFR are co-endocytosed with E-cad, promoting non-canonical EGFR:STAT signalling. Elevated activity of this signalling caused by increased intracellular accumulation of E-cad, leads to the activation of the JNK pathway and apoptosis. At the same time, increased intracellular accumulation of E-cad also reduces HP1 localisation to the chromatin, suggesting that the

activation of EGFR:STAT signalling outcompetes STAT from HP1. Altogether, our work provides evidence that the compartmentalisation of STAT by endocytosis determines the signalling outputs.

## Results

To investigate the cellular and molecular functions of non-canonical STAT signalling, we utilised the *Drosophila* STAT92E variant with the substitution of the conserved tyrosine residue in position 704 for phenylalanine (Y704F). This variant cannot be phosphorylated by JAK and is therefore only able to function non-canonically [38]. To gain an insight into potential transcriptional changes mediated by non-canonical STAT activities, we performed mRNA sequencing from L3 stage larval wing discs either overexpressing a short hairpin RNA designed to act as an siRNA targeting HP1 (*HP1* RNAi), overexpressing STAT92E$^{Y704F}$ or expressing myr::GFP (control) for 24 hours (see Materials and Methods). Using a statistical cutoff adjusted $p < 0.05$ (see Materials and Methods), we found that downregulation of HP1 ($log_2$ fold change -100.1, adjusted $p < 5E-324$) changed the expression of 1876 genes, whereas overexpression of STAT92E$^{Y704F}$ ($log_2$ fold change 28.0, adjusted $p = 5.1E-168$) affected the expression of 4120 genes. Remarkably, 889 genes changed expression in both datasets ($p = 2.4e-69$, Fig 1A), indicating that 47% of genes affected by *HP1* RNAi also appear to be regulated by non-canonical STAT signalling. Moreover, the direction of change was consistent for 700 of these 889 genes (79%), with genes being both upregulated and downregulated (Fig 1B, numerical data for this and all the following plots is in S1 Table) and a strong correlation between both datasets (Fig 1B, $r^2 = 0.51$, $p < 2.2e-16$).

The similar changes in gene expression patterns following STAT92E$^{Y704F}$ overexpression and HP1 knockdown made us hypothesise that non-canonical STAT92E exerts an effect that HP1 normally inhibits in epithelial wing disc cells. To investigate the nature of such a function, we examined the localisation of STAT::GFP – STAT tagged via the insertion of GFP at its C-terminal end, expressed under the control of its genomic region and able to rescue the lethality and fertility of *Stat92E* mutant flies [18]. Spatially, we focused on the dorsal wing pouch, where the activity of canonical JAK/STAT signalling is low/undetectable (Fig 1C), as visualised with an activity reporter containing multiple canonical STAT92E DNA binding sites (*10xSTAT92E*-GFP) [39]. As previously described, control wing disc cells contained STAT::GFP in the cytosol and nucleus [40], however, we also detected STAT::GFP at cell-cell boundaries, where it co-localised with E-cad (Fig 1D and 1E).

As STAT92E and E-cad colocalise, and the Par-3 has previously been shown to bind to both [18,19], we asked if the localisation of STAT::GFP at cell-cell boundaries depends on E-cad. To manipulate E-cad levels, we overexpressed untagged full-length E-cad throughout the developing wing pouch [41]. Such an overexpression moderately increased E-cad levels at cell-cell borders (Figs 1D and S1A) and substantially increased levels in the cytoplasm, where E-cad also appears to be enriched within intracellular vesicles (Fig 1F). This relative redistribution was reflected by the increase in the ratio of E-cad in the cytoplasm relative to cell-cell borders (Fig 1G). We found that the localisation of STAT::GFP at cell-cell borders was largely lost upon E-cad overexpression (Fig 1D, 1F and 1G), a change that was resulted in an increased STAT::GFP ratio in the cytoplasm relative to cell borders (Fig 1G), similar to that observed for E-cad. In addition, STAT::GFP and E-cad were detected colocalised in vesicles following E-cad overexpression (Fig 1F). Based on these findings, we suggest that E-cad recruits STAT92E to cell-cell borders, potentially via Par-3, from where it co-traffics with STAT92E in intracellular vesicles upon E-cad overexpression.

We next sought to determine how the relocalisation of STAT::GFP by E-cad overexpression might alter its signalling. We used the *ptc*-Gal4 driver expressed in cells along the anterior-posterior compartment boundary in the wing discs [42,43], to allow the direct comparison of neighbouring cells with either normal or increased E-cad levels (Fig 1C). Intriguingly, E-cad overexpression did not affect the expression of the canonical activity reporter *10xSTAT92E*-GFP in either the wing disc pouch or the dorsal hinge region (Figs 1H and S1B) – areas with low and high canonical JAK/STAT signalling respectively (Fig 1C). Therefore, if E-cad has any effects on STAT92E activity, then this is likely to be through non-canonical signalling pathways. As there are no reporters for non-canonical STAT92E signalling, we sought to analyse

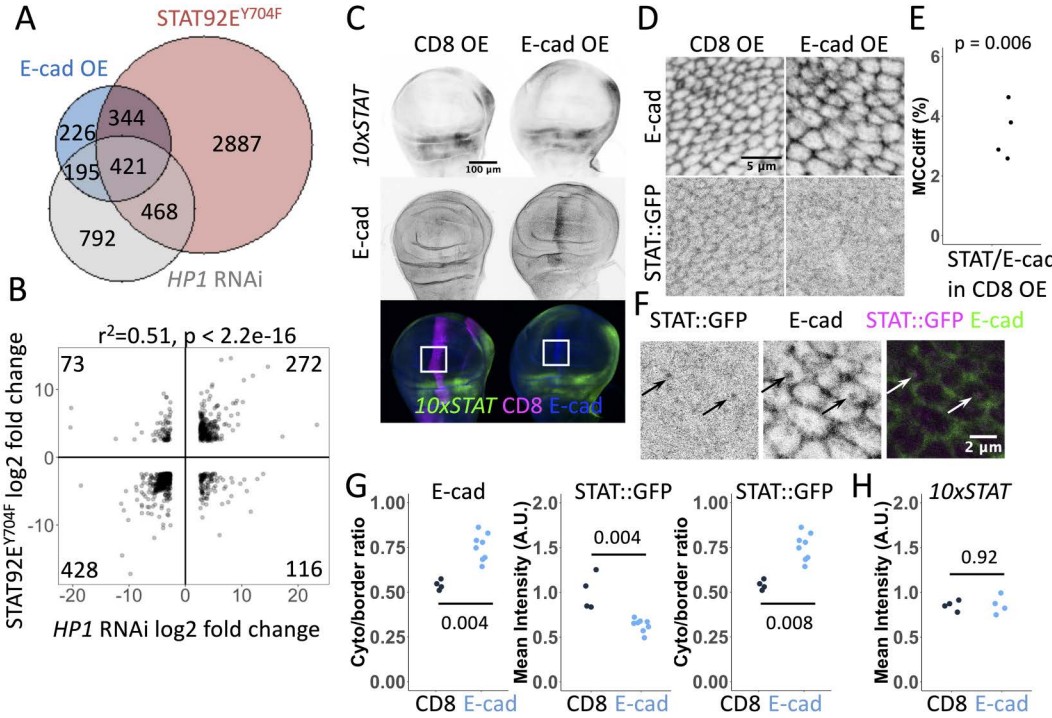

**Fig 1. Gene expression changes by non-canonical STAT92E signalling and the control of STAT92E localisation by E-cadherin.** (**A**) Scaled Venn diagram depicting genes with significantly changed expression (p < 0.05) in wing discs upon overexpression of E-cad::EOS and STAT92E^Y704F and *HP1*-RNAi for 24 hours using *Act5C*-Gal4. (**B**) Correlation of gene expression changes in wing discs upon overexpression of STAT92E^Y704F and *HP1* knockdown. Each dot represents an individual gene. (**C**) Representative images *10xSTAT92E*-GFP (*10xSTAT*) expression in wing discs expressing CD8::mCherry (CD8 OE, left) and full-legth untagged E-cad (E-cad OE, right). CD8::mCherry (bottom left, magenta) and E-cad (bottom right, blue) were expressed using *ptc*-Gal4 and the levels of GFP (direct fluorescence, top row, grey; bottom row, green) were measured within the expressing stripe in the wing pouch relative to the outside of the stripe. White squares outline the dorsal wing pouch area used for quantifications in this and other experiments. (**D**) Representative images of STAT::GFP localisation (bottom row) expressed at the endogenous level in wing discs overexpressing CD8::mCherry (CD8, left) and full-legth untagged E-cad (right). E-cad staining by antibody is in the top row. *MS1096*-Gal4 driver was used. (**E**) Mander's colocalisation coefficient (MCC_diff, see Materials and Methods) for the co-localisation of STAT::GFP with E-cad signal (N = 4, sections = 8 per sample). One-sample t-test in comparison to zero for random colocalisation was used. (**F**) A single confocal section depicting STAT::GFP expressed at the endogenous level (left, grey; right, magenta) and E-cad (antibody staining, middle, grey; right, green) in wing discs overexpressing E-cad. Arrows highlight examples of their colocalisation in intracellular puncta. (**G**) Quantification of ratios between the cytoplasm and cell-cell border levels of E-cad (left) and STAT::GFP (right) and mean STAT::GFP levels at cell-cell borders in wing discs expressing CD8 (left, black) and E-cad (right, blue). The Wilcoxon test was used to compare the datasets. N = 4 and 8 wing discs. (**H**) Quantification of *10xSTAT92E*-GFP levels (*10xSTAT*) in the wing pouch area of wing discs expressing CD8::mCherry and E-cad as shown in (**C**). The Wilcoxon test was used to compare the datasets. N = 4 and 4 wing discs. The quantification of levels in the notum is in S1B Fig.

the mRNA transcriptome as a way to identify dysregulated pathways at the whole genome level. To this end, we overexpressed E-cad tagged with monomeric EOS fluorescent protein (E-cad::EOS) [44,45], a transgene that increases E-cad protein levels both at cell-cell borders and in the cytoplasm (S2A and S2B Fig). We found that overexpression of E-cad::EOS (log$_2$ fold change 19.3, adjusted p = 2E-79) changed the expression of 1186 genes, of which 765 genes were also affected by STAT92E^Y704F overexpression (p < 8.552e-151, Fig 1A). Therefore, nearly 65% of genes affected by E-cad overexpression are also regulated by non-canonical STAT92E signalling. Out of these 765 genes, the direction of change was consistent for 738 genes (96.5%, Fig 2A, a correlation coefficient r$^2$ = 0.85, p < 2.2e-16). Furthermore, comparing the E-cad dataset to our original data (Fig 1A), we identified 421 loci (55% of those affected by both E-cad and STAT92E^Y704F, p = 4.5e-177), which also changed their expression following HP1 knockdown (Fig 1A).

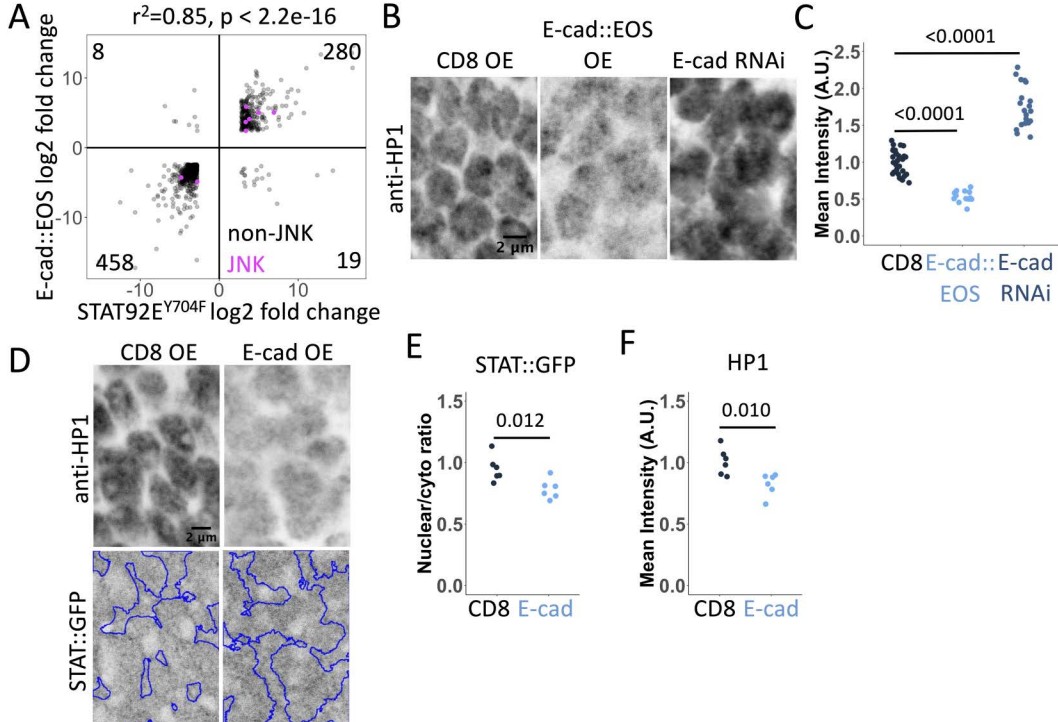

**Fig 2. Changes in gene expression, HP1 and STAT::GFP localisation following E-cadherin overexpression.** (**A**) Correlation of gene expression changes in wing discs upon overexpression of E-cad::EOS and STAT92$^{EY704F}$. Each dot represents an individual gene. Dots in magenta represent components of the JNK pathway. (**B-C**) Representative images (B) and quantification (C) of HP1 levels (antibody staining) in wing discs with E-cad::EOS overexpression (OE, middle), E-cad knockdown (RNAi, right) or overexpression of CD8::mCherry as control (left) for 24 hours using *ptc*-Gal4. Each dot represents a single nucleus. One-way ANOVA and post-hoc t-test with false discovery rate p-value correction were used. N = 29/3, 16/3 and 20/4 (nucleus/wing disc). Localisation of E-cad::EOS is shown in S2 Fig. Similar data for H3K9me3 is in S3 Fig. (**D-F**) Representative images (D) and quantification (E-F) of STAT::GFP (bottom in D, E) and HP1 (antibody staining, top in D, F) levels in wing discs with overexpression of the untagged full-length E-cad (right) or CD8::mCherry as control (left) for 24 hours using *ptc*-Gal4. Blue outline in D represents the mask based on the HP1 signal in the areas not expressing transgenes in the same wing discs. Each dot represents the mean intensity in individual wing discs. The Wilcoxon test was used to compare the datasets. N = 6 and 6.

Given the overlap in gene expression changes identified following E-cad overexpression and HP1 knockdown, we next asked if E-cad overexpression and its recruitment of STAT92E to intracellular vesicles might interfere with the recruitment of STAT92E to HP1 and, thus, the chromatin stabilisation activity of non-canonical unphosphorylated STAT92E. To explore this, we tested how changing E-cad expression affects HP1 levels in the nuclei of wing disc cells. We found that E-cad::EOS overexpression reduced the average HP1 levels, whereas E-cad knockdown had the opposite effect and increased them (Fig 2B and 2C). These changes in HP1 levels were consistent with the changes in H3K9me3 staining – the modified Histone 3 bound by HP1 – in the same conditions (S3A and S3B Fig), a finding that aligns with the known HP1 and non-canonical STAT requirement for this histone mark [5,46]. To test whether the altered HP1 nuclear levels correlate with STAT nuclear accumulation, we measured the effects of overexpressing the untagged full-length E-cad on the STAT::GFP nuclei/cytoplasm ratio in the wing pouch, where the activity of the canonical JAK/STAT signalling is low. Overexpression of untagged E-cad reduced HP1 nuclear levels and nuclear/cytoplasm STAT::GFP ratio (Fig 2D and 2F). These data are consistent with a model in which E-cad in the cytosol sequesters nuclear unphosphorylated STAT92E away from HP1, suggesting that STAT92E fulfils two distinct non-canonical functions – nuclear and cytosolic.

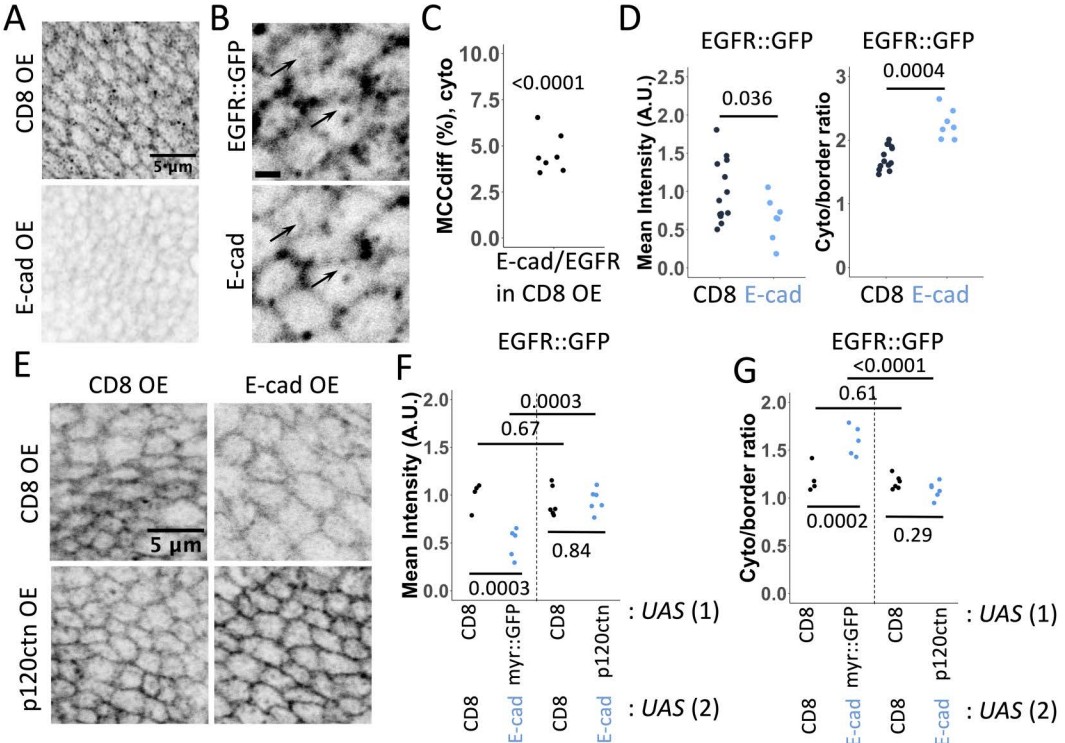

**Fig 3. Effects of E-cadherin overexpression and endocytosis on EGFR::GFP localisation.** (**A**) Representative images of EGFR::GFP expressed from the endogenous locus in wing discs overexpressing CD8::mCherry (CD8 OE, top) and E-cad (E-cadOE, bottom). (**B**) A single confocal section depicting EGFR::GFP expressed from the endogenous locus (left) and E-cad (antibody staining, right). Arrows highlight examples of their colocalisation on vesicles. Scale bar 1 μm. (**C**) Mander's Colocalisation Coefficient (MCC$_{diff}$, cyto) for the colocalisation of E-cad with EGFR::GFP in cytoplasm. p-value shows the comparison of MCC$_{diff}$ with zero (= no colocalisation) using a one-sample t-test (N = 7, sections = 8 per sample). (**D**) Quantification of mean levels at cell-cell borders (left) and the ratio between the cytoplasm and cell-cell border levels (right) of EGFR::GFP expressed from the endogenous locus in control wing discs (left, black) and those overexpressing E-cad (right, blue). The Wilcoxon test was used to compare the datasets. N = 13 and 7 wing discs. (**E-G**) Representative images (E) and quantification of ratios between the cytoplasm and cell-cell border levels of EGFR::GFP (F) and mean EGFR::GFP levels at cell-cell borders (G) in wing discs co-expressing CD8::mCherry (CD8, black, left in E) and E-cad (blue, right in E) with either CD8::mCherry (CD8, top in E) or p120-catenin (p120ctn, bottom in E). Two-way ANOVA and post-hoc t-test with false discovery rate p-value correction were used. p = 0.0002 (F) and p = 0.0009 (G) for the interaction between E-cad and p120-catenin overexpression. N = 4, 5, 6, 6 left-to-right. The *MS1096*-Gal4 driver was used in A-G.

Since EGFR was linked with both E-cad and non-canonical STAT signalling [8–10,21], we next investigated the potential role of EGFR in cytosolic, non-canonical STAT92E activity. We found that an EGFR::GFP fusion protein expressed from the endogenous EGFR locus colocalised with E-cad at both the cell-cell borders and intracellular vesicles in wing disc cells (Fig 3A–3D). Furthermore, overexpression of E-cad exerted a similar effect on EGFR::GFP as it did to STAT::GFP, reducing levels of EGFR::GFP at cell-cell borders with a concomitant increase in the ratio of EGFR::GFP in the cytoplasm (Fig 3A and 3D). To test whether EGFR::GFP redistribution upon E-cad overexpression was caused by E-cad endocytosis from the plasma membrane, we additionally overexpressed p120-catenin, a treatment previously shown to block E-cad endocytosis [34]. As anticipated, inhibiting E-cad endocytosis restored EGFR::GFP levels at the plasma membrane and the ratio of EGFR::GFP in the cytoplasm (Fig 3E–3G), supporting a model in which E-cad endocytosis is directly required for EGFR relocalisation.

The colocalisation of both EGFR and STAT92E at cell-cell borders and their similar redistribution following E-cad overexpression suggested that E-cad, EGFR and STAT92E intersect in a single pathway. To explore this possibility, we

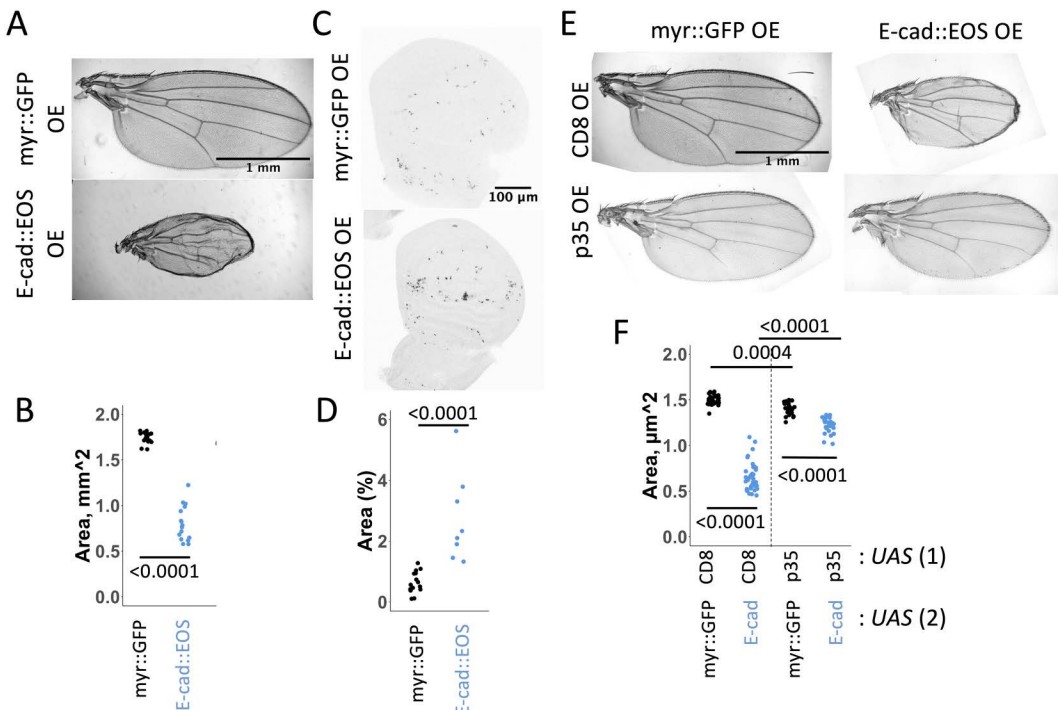

**Fig 4. E-cadherin overexpression reduces adult wing size by inducing apoptosis.** (A-B) Representative image of adult wings (A) and Dcp-1 staining in wing imaginal discs (B) expressing either myr::GFP (top) or E-cad::EOS (bottom). (C-D) Quantifications of adult wing area (B) and Dcp-1 staining (% area, D) in wing discs expressing myr::GFP (left) or E-cad::EOS (right). The Wilcoxon test was used to compare the datasets. N = 17 and 16 (C), 15 and 7 (D). (E-F) Representative images (E) and area quantification of adult wings expressing either myr::GFP (F, left column, B, black) or E-cad::EOS (E, right column, F, blue) together with either CD8::mCherry (CD8, E, top row, F, left) or p35 (E, bottom row, F, right). Two-way ANOVA and post-hoc t-test with false discovery rate p-value correction were used. N = 31, 30, 37 and 28, left-to-right. p < 0.0001 for the interaction between E-cad::EOS and p35 overexpression.

focused on apoptosis as a functional output, as both STAT activation by EGFR and intracellular E-cad accumulation have previously been associated with this process [8–10,36]. Overexpression of both E-cad::EOS and untagged E-cad reduced the size of the resulting adult wing (Figs 4A, 4B, and S4A) and increased the amount of apoptosis in wing discs as shown by an antibody detecting cleaved *Drosophila* death caspase-1 (Dcp-1, Figs 4C, 4D, S4B and S4C). To confirm that the small adult wing size caused by E-cad overexpression resulted from increased apoptosis, we co-expressed E-cad with the apoptosis-blocking baculoviral p35 protein [47]. Co-expression of p35 rescued the small adult wing size caused by E-cad overexpression (Fig 4E and 4F), demonstrating the causal relationship between adult wing size and apoptosis and supporting the use of adult wing size as a reliable readout for levels of apoptosis in larval wing discs, in agreement with our previous work [36].

Using adult wing size as a proxy for levels of cell death in larval wing discs, we found that halving the gene dose of *Egfr* was sufficient to rescue the small adult wing size caused by E-cad overexpression and reduce the number of apoptotic cells detected in larval wing discs (Fig 5A–5D). To exclude the possibility that this rescue was due to a reduction in E-cad protein levels rather than downstream signalling by EGFR, we measured potential changes in E-cad levels in heterozygous *Egfr* individuals and following expression of a dominant-negative EGFR variant. Neither of these manipulations altered E-cad levels at cell-cell borders or the cytoplasm to cell-cell border ratio of E-cad (Figs 5E, 5F, S4D and S4E). Taken together, these findings are consistent with the hypothesis that E-cad and EGFR act in the same pathway to promote apoptosis.

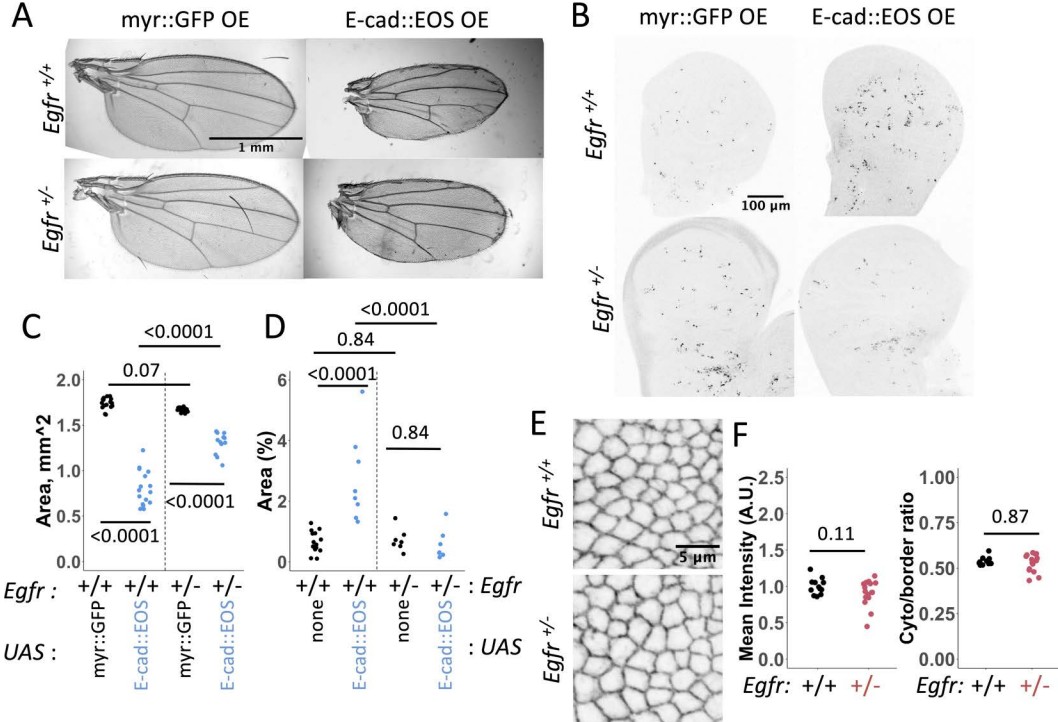

**Fig 5. EGFR acts downstream of E-cadherin in inducing apoptosis.** (**A-B**). Representative images (A) and area quantification (B) of adult wings expressing either myr::GFP (A, left column, B, black) or E-cad::EOS (A, right column, B, blue) in the presence of two functional copies (A, top row, B, left) or one copy (A, bottom row, B, right) of the *Egfr* gene. (**C-D**) Representative images (C) and quantification (D) of Dcp-1 staining in L3 larval wing discs expressing myr::GFP (C, left column,D, black) or E-cad::EOS (C, right column, D, blue) in the presence of two functional copies (C, top row, D, left) or one copy (C, bottom row, D, right) of the *Egfr* gene. Two-way ANOVA and post-hoc t-test with false discovery rate p-value correction were used. p<0.0001 for the interaction between E-cad overexpression and *Egfr* copy number in both B and D. N = 17, 16, 17 and 12, left-to-right (B) and = 15, 7, 8 and 7, left-to-right (D). Data for the overexpression of untagged E-cad is in S4 Fig. (**E-F**) Representative images (E) and quantification (F) of mean cell-cell border levels (F, left) and the ratio between the cytoplasm and cell-cell border levels (F, right) of E-cad::GFP expressed from the endogenous locus in the presence of two functional copies (E, left, F, black) or one copy (E, right, F, red) of the *Egfr* gene. The Wilcoxon test was used to compare the datasets. N = 12 and 16 wing discs. Similar data for the expression of the dominant-negative variant of EGFR is in S4 Fig.

To further test whether STAT92E is involved in the same pathway as EGFR and E-cad, we reduced the gene dose of *stat92E.* Heterozygosity for *stat92E* mutant allele suppressed the adult wing phenotypes caused by E-cad overexpression (Fig 6A and 6B), suggesting that STAT92E is also necessary for the induction of apoptosis in an E-cad overexpression background. We confirmed the specificity of EGFR and STAT92E in this phenotype by reducing the doses of genes involved in the Wnt and Hippo pathways previously shown to be regulated by E-cad [15,48]. Targeting these pathways did not affect the adult wing size phenotype (S5A and S5B Fig). In agreement with the hypothesis that EGFR acts via STAT92E to promote apoptosis, we did not see any changes in the Ras/MAPK pathway activity following manipulation of E-cad levels as visualised by antibodies against phosphorylated ERK (S4F and S4G Fig).

Reducing the gene dose of *hopscotch* (*hop*) – the gene encoding the single *Drosophila* JAK protein – did not modulate adult wing size following E-cad overexpression (S5A and S5B Fig), further supporting our findings that E-cad overexpression does not affect the canonical JAK/STAT signalling but interferes with non-canonical STAT92E functions. To investigate the hypothesis that there are two competing non-canonical STAT92E pathways – nuclear and cytoplasmic – we knocked down HP1 using RNAi, producing a dose-dependent reduced adult wing size phenotype similar to those of E-cad overexpression (Fig 6C and 6D). Moreover, the simultaneous overexpression of the untagged full-length E-cad and

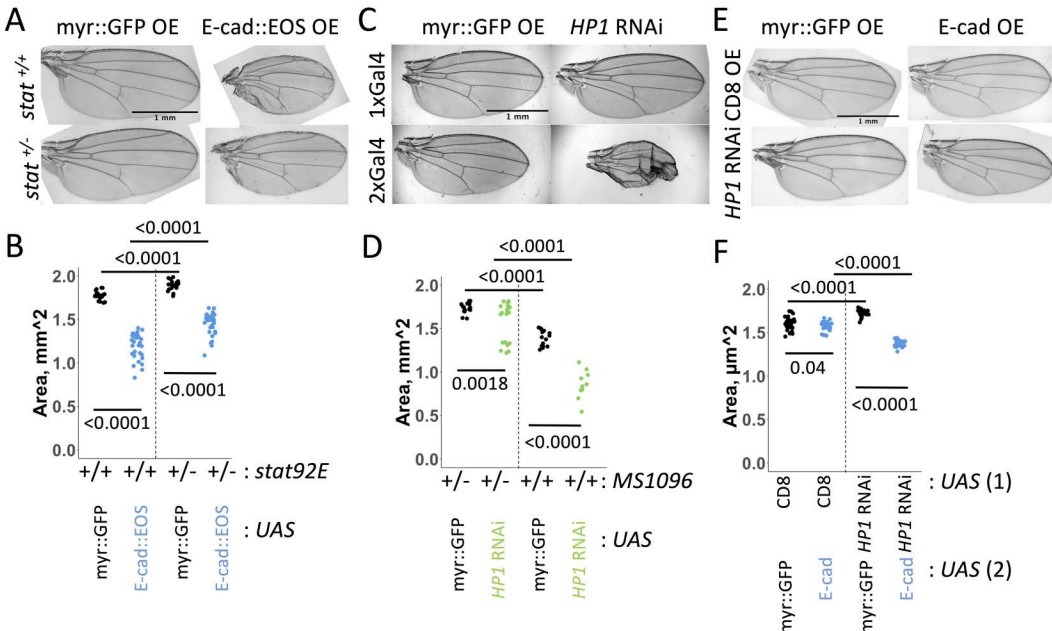

**Fig 6. E-cadherin interacts with STAT92E and HP1 in determining adult wing size. (A-B)** Representative images (A) and area quantification (B) of adult wings expressing either myr::GFP (A, left column, B, black) or E-cad::EOS (A, right column, B, blue) in the presence of two functional copies (A, top row, B, left) or one copy (A, bottom row, B, right) of the *stat92E* gene. Two-way ANOVA and post-hoc t-test with false discovery rate p-value correction were used. N = 21, 32, 34 and 33, left-to-right. p = 0.02 for the interaction between E-cad::EOS overexpression and *stat92E* copy number. The lack of rescue by reducing doses of components of Wnt and Hippo pathways as well as reducing the gene dose of *hopscotch* (*hop*) encoding for the single *Drosophila* JAK protein is shown in S5 Fig. **(C-D)** Representative images (C) and area quantification (D) of adult wings expressing myr::GFP (C, left, D, black) or *HP1* RNAi (C, right, D, green) using one copy of *MS1096*-Gal4 (C, top, D, left) or two copies (C, bottom, D, right). Two-way ANOVA and post-hoc t-test with false discovery rate p-value correction were used. N = 17, 19, 15 and 10, left-to-right. p < 0.0001 for the interaction between the presence of *HP1* RNAi and *MS1096*-Gal4 copy number. **(E-F)** Representative images (E) and area quantification (F) of adult wings expressing either myr::GFP (E, left column, F, black) or E-cad::EOS (E, right column, F, blue) together with either CD8::mCherry (CD8, E, top row, F, left) or *HP1* RNAi (E, bottom row, F, right). Two-way ANOVA and post-hoc t-test with false discovery rate p-value correction were used. N = 34, 23, 25 and 23, left-to-right. p < 0.0001 for the interaction between E-cad::EOS overexpression and *HP1* knockdown.

*HP1* RNAi combined to synergistically reduce the adult wing size (Fig 6E and 6F), further supporting two competing non-canonical STAT92E pathways.

Next, we tested the contribution of E-cad endocytosis to cell death, given that elevated E-cad endocytosis that led to E-cad intracellular accumulation was required to induce apoptosis following AP-1 knockdown [36]. Consistent with this, we found that co-overexpression of p120-catenin partly rescued the adult wing size phenotype caused by E-cad overexpression (Fig 7A and 7B). Taken together with the observed colocalisation of E-cad with STAT92E and EGFR on intracellular vesicles (Figs 1F and 2E) and the rescue of EGFR::GFP cell-cell border localisation by co-overexpressing p120-catenin and E-cad (Fig 3E and 3F), we suggest that co-endocytosis of E-cad with STAT92E and EGFR promotes the activation of the EGFR:STAT signalling and apoptosis.

Finally, we asked if the changes in gene expression observed upon overexpression of E-cad and STAT92E^EY704F could represent 'effector genes' whose changed expression might mediate the cell death phenotype caused by E-cad overexpression. Examining the relative enrichment of GO terms within our transcriptomics dataset, we found that genes involved in the JNK cascade were over-enriched in the group of transcripts positively regulated by E-cad and STAT92E^Y704F overexpression (Fig 2A, p = 0.006), with six genes out of 82 known components of the *Drosophila* JNK pathway increased in both datasets, 23 in the STAT92E and 9 in the E-cad datasets. Of the six genes – *mitogen-activated protein kinase kinase kinase* (*Mekk1*),

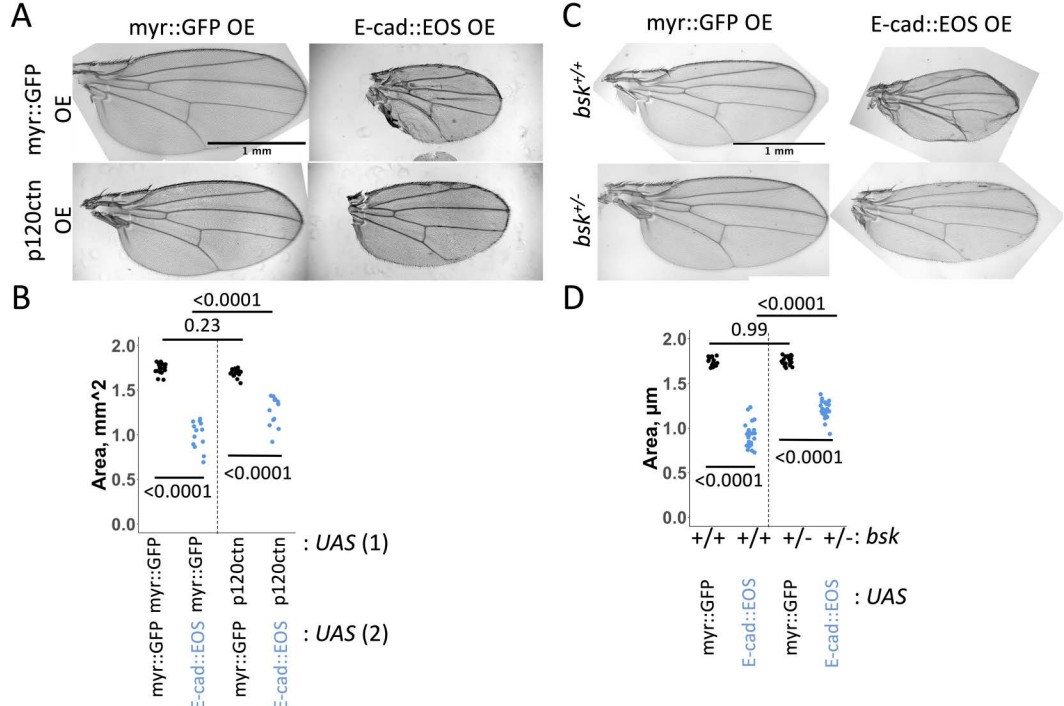

**Fig 7. E-cadherin endocytosis and JNK signalling contribute to the small wing phenotype caused by E-cadherin overexpression. (A-B)** Representative images (C) and area quantification of adult wings expressing either myr::GFP (C, left column, D, black) or E-cad::EOS (C, right column, D, blue) together with myr::GFP (C, top row, D, left) or p120-catenin (C, bottom row, D, right, p120ctn) using the *MS1096*-Gal4 driver. Two-way ANOVA and post-hoc t-test with false discovery rate p-value correction were used in D. N = 17, 13, 16 and 12, left-to-right. p < 0.0001 for the interaction between E-cad::EOS and p120ctn overexpression. **(C-D)** Representative images (C) and area quantification of adult wings expressing either myr::GFP (C, left column, D, black) or E-cad::EOS (C, right column, D, blue) in the presence of two functional copies (C, top row, D, left) or one copy (C, bottom row, D, right) of the *basket* (*bsk*) gene. Two-way ANOVA and post-hoc t-test with false discovery rate p-value correction were used. N = 15, 20, 26 and 22, left-to-right. p < 0.0001 for the interaction between E-cad::EOS overexpression and *basket* copy number.

*Plenty of SH3s* (*POSH*), *spoonbill* (*spoon*), *Growth arrest and DNA damage-inducible 45* (*Gadd45*), *Autophagy-related 9* (*Atg9*), and *fiery mountain* (*fmt*) – five have been described as positive regulators of JNK signalling. Given this strong indication of JNK pathway involvement, we therefore examined whether increased JNK pathway activity could explain increased apoptosis. We tested a mutant allele of *basket* (*bsk*), encoding the only *Drosophila* JNK. Reducing the gene dose of *bsk* in an E-cad::EOS overexpressing background improved the adult wing phenotype (Fig 7C and 7D). In addition, we also tested *POSH*, whose expression was increased in all three RNAseq datasets and whose ectopic expression has previously been shown to be sufficient for JNK activation [49]. As in the case of *bsk*, reducing the gene dose of *POSH* in an E-cad overexpressing background rescued the adult wing phenotype, but also the amount of apoptosis in larval wing discs (Fig 8A–8D). Taken together, these results support the conclusion that JNK pathway activity is necessary for apoptosis induction by elevated E-cad and EGFR:STAT signalling.

## Discussion

In this work, we provide novel, in vivo evidence identifying the biological functions of non-canonical STAT92E activity in *Drosophila* wing development. We demonstrate that these non-canonical roles depend on E-cad, EGFR and JNK-pathway activation (Fig 9) – the first time the trans-membrane adhesion protein E-cad has been demonstrated to play a part in the non-canonical STAT activities. The activation of these non-canonical STAT92E pathways is most clearly demonstrated

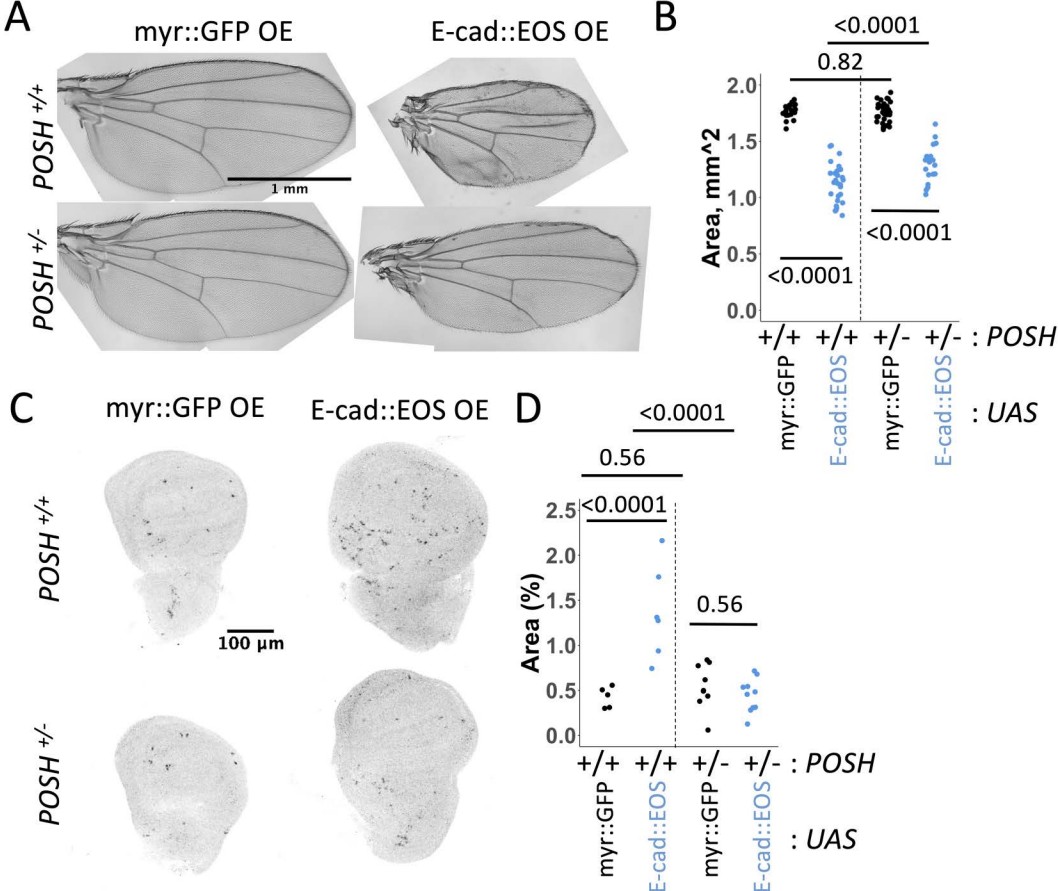

**Fig 8. *POSH* acts downstream of E-cadherin in inducing apoptosis.** (**A-B**) Representative images (A) and area quantification of adult wings expressing either myr::GFP (A, left column, B, black) or E-cad::EOS (A, right column, B, blue) in the presence of two functional copies (A, top row, B, left) or one copy (A, bottom row, B, right) of the *POSH* gene. Two-way ANOVA and post-hoc t-test with false discovery rate p-value correction were used. N = 18, 27, 33 and 20, left-to-right. p = 0.001 for the interaction between E-cad::EOS overexpression and *POSH* copy number. (**C-D**) Representative images (C) and quantification (D) of Dcp-1 staining in wing discs expressing myr::GFP (left in C, black in D) or E-cad::EOS (right in C, blue in D) in the presence of two or one copies of the *POSH* gene. Two-way ANOVA and post-hoc t-test with false discovery rate p-value correction were used. p < 0.0001 for the interaction between E-cad overexpression and *POSH* copy number. N = 5, 9, 6 and 10, left-to-right.

by the striking overlap in the patterns of gene expression changes mediated by STAT92E$^{Y704F}$ and E-cad overexpression as well as by the knockdown of HP1 (Figs 1A and 2A), with the co-localisation of STAT92E and EGFR with E-cad supporting a model, in which a complex between them plays a role in this novel pathway. In addition, we demonstrate the involvement of EGFR and STAT92E in mediating the small wing size phenotype following E-cad overexpression and endocytosis. Finally, we show that this small wing phenotype is mediated by the transcriptional control of JNK-pathway regulators, which activate signalling to induce apoptosis. While not previously considered together, STAT-mediated effects downstream of EGFR signalling [11], physical interactions between E-cad and STAT92E [20], and the interaction between unphosphorylated STAT92E and HP1 in the regulation of heterochromatinisation [5,50] have each been individually described before and support the findings presented here.

One key aspect of our study is the interaction of endocytic trafficking of E-cad, STAT92E and EGFR. While the nature of the E-cad, EGFR and STAT92E-containing vesicles and their fate are yet to be elucidated in vivo, previous reports have demonstrated the importance of endocytosis for both STAT92E [24,26] and E-cad [31,36] functions. We hypothesise

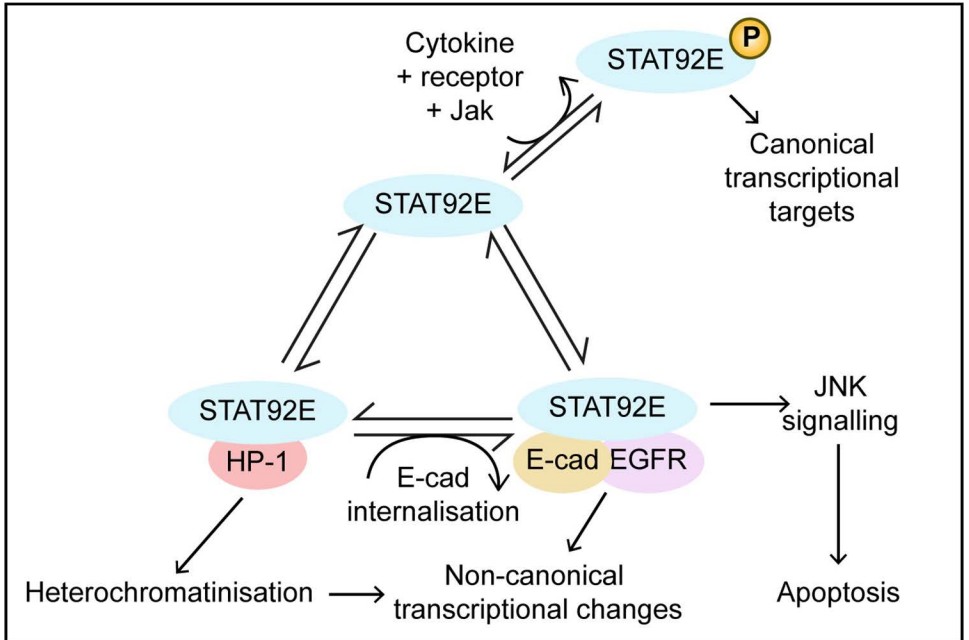

**Fig 9. The model of STAT activities compartmentalisation in a cell.** Three distinct modalities of STAT signalling co-exist in a cell: the canonical JAK/ STAT pathway and the non-canonical EGFR:STAT and HP1:STAT pathways. The JAK/STAT pathway is regulated by endocytosis, as previously shown [24] and involves STAT phosphorylation on the Y704 residue. Cytosolic non-phosphorylated STAT is available for nuclear translocation and HP1 interaction. Non-phosphorylated STAT is also recruited from the cytosol to E-cad:EGFR at the plasma membrane and intracellular vesicles. In the scenario of the elevated intracellular E-cad, high levels of EGFR and STAT are recruited to the E-cad containing vesicles. Elevated EGFR:STAT signalling triggers apoptosis through upregulation of the JNK pathway. At the same time, STAT outcompetes HP1 binding, thus reducing HP1 levels on DNA.

that these vesicles are destined for lysosomal degradation, which likely explains the apparent decrease in EGFR and STAT92E at the plasma membrane upon E-cad overexpression (see Figs 1D and 3). In addition, the requirement for E-cad endocytosis indicates that specific endocytic compartments may facilitate the EGFR:STAT signalling activity. For example, the tyrosine kinase PYK2 is phosphorylated and recruited to early endosomes by EGFR, where it colocalises with EGFR and sustains EGFR:STAT3 signalling in human mammary MCF10A and MDA-MB-468 cells [30]. As such, regulation of an E-cad:EGFR:STAT92E complex within a particular endocytic compartment would not be an exception. Indeed, the assembly of specific signalling complexes on early endosomes to quantitatively and qualitatively alter signalling outputs is an established paradigm [51]. Furthermore, endocytosis appears to be an important regulator of not only non-canonical STAT signalling as demonstrated in this work, but also canonical JAK/STAT signalling [24]. In the latter case, the activated receptor needs to reach early endosomes to activate the *10xSTAT92E*-Luciferase activity reporter, which utilises the same promoter as the *10xSTAT92*-GFP used in this work. In contrast, the expression of other pathway targets – such as *socs36E* mRNA – was not achieved at early endosomes but rather required further progression of trafficking to late endosomes [24]. Taken together, these findings support the existence of specialised endosomes, or 'signalosomes' that enable distinct signalling activities via both canonical and non-canonical pathways.

The outcome of many signalling pathways is a change in gene expression. In this work, we identified the extensive overlap in the altered gene expression patterns elicited by manipulations of STAT92E, E-cad and HP1 levels. While we have deposited these data sets in publicly available databases (https://www.ncbi.nlm.nih.gov/bioproject/1185032) to allow future analysis, in this report, we have focused specifically on genes encoding components of the JNK pathway. Taken together with previous reports linking elevated intracellular E-cad levels to activation of the JNK pathway [36], interactions

with the JKN-kinase *basket* (Fig 7C and 7D) and the upregulation of pathway activators such as *POSH* – able to rescue of apoptosis in larval wing discs and the small wing phenotype (Fig 8) – all support the view that the E-cad:EGFR:STAT92E signalling promotes apoptosis via JNK pathway activity induction (Fig 9). Additionally, we observed a much greater number of genes changing their expression following modulation of non-canonical STAT or HP1 levels than has been reported in previous experiments using the human colon cancer (DLD-1) cell line [6]. While we observed 4120 genes with altered expression upon STAT92E$^{Y704F}$ overexpression (31.8% of all protein-encoding genes), only 162 genes (0.32% of 47,324 transcripts on the Illumina Human gene chip) displayed altered expression following overexpression of non-canonical STAT5A[7]. Such a discrepancy could be due to differences in overexpression levels, compensation/redundancy by other STATs in mammalian cells or specific cell types used. It is worth noting that the previous work used DLD-1 cells, which have very low basal levels of EGFR [6,52] and so are unlikely to have the non-canonical EGFR:STAT activity. Therefore, our data suggest that the non-canonical STAT92E signalling has a major role in regulating the gene expression landscape in epithelial cells in *Drosophila*, likely through interactions with both EGFR and HP1.

Several lines of evidence suggest that non-canonical STAT92E activities compete for unphosphorylated STAT92E availability in wing disc cells. For example, modulations of E-cad levels were sufficient to change HP1 present in nuclei and STAT nuclear/cytoplasm ratios (Fig 2), suggesting that E-cad is able to outcompete STAT92E from HP1 complexes. Consistent with this, E-cad-induced wing size decreases were suppressed by reducing STAT92E levels (Fig 6A and 6B), indicating that STAT92E is rate-limiting for triggering apoptosis in this tissue. Furthermore, expressing two copies of *HP1* RNAi or co-expressing untagged full-length E-cad with one copy of *HP1* RNAi also led to a decrease in adult wing size (Fig 6C–6F), further supporting a model where HP1 and E-cad mutually compete for the available STAT92E pool. However, overexpression of E-cad did not alter canonical STAT92E signalling as assessed by the transcriptional reporter *10xSTAT92E*-GFP (Fig 1C). Therefore, we propose that the canonical JAK/STAT92E signalling is dominant over the two other modalities and outcompetes STAT92E from both HP1 and EGFR. This view is supported by the observation that the gain-of-function *Tumerous-lethal (Tuml)* allele of *Drosophila* JAK inhibits heterochromatin formation and reduces HP1:STAT association [5,53] – a characteristic also shared by the gain-of-function JAK2 V617F allele associated with the majority of human myeloproliferative neoplasms [54].

Considering our results in a broader context, we propose that the apoptosis induced by E-cad endocytosis and EGFR:STAT92E activation provides a mechanism for the elimination of potential cancer cells undergoing cell-cell adhesion disassembly during epithelial-to-mesenchymal transitions and, thus, is likely to play an important tumour-suppressive role. While the disassembly of E-cad cell-cell adhesion is a known trigger of apoptosis [55–57], the molecular basis of this induction was not previously known. However, it should be noted that blocking the apoptosis induced by E-cad endocytosis and EGFR:STAT92E activation was insufficient in itself to drive tumorigenic transformation (Fig 4E and 4F). Therefore, we propose that the mechanism of apoptosis induction by E-cad endocytosis represents just one of the steps that an epithelial cell needs to overcome to become cancerous, in agreement with the evolutionary model of multistage carcinogenesis [58,59]. At the same time, there must be mechanisms in place that inhibit the undesired induction of apoptosis. For example, epithelial-to-mesenchymal transitions that occur during midgut formation in early *Drosophila* embryos require adhesion disassembly and E-cad endocytosis but are not accompanied by cell death [60]. Discovering these inhibitory mechanisms is an important next step, as they are likely to present oncogenic pathways that induce apoptosis resistance and help cancer cells avoid death.

In summary, we report on two competing non-canonical STAT92E pathways which concurrently act in the epithelial cells of *Drosophila* wing imaginal discs. One pathway is linked to HP-1 and stabilises heterochromatin, while the other is dependent on EGFR to promote apoptosis via the activation of JNK pathway signalling and requires E-cad endocytosis for its activation. The validity of our findings discussed here is further supported by mirrored activities in vertebrate systems. While the binding between E-cad and EGFR is long established [14], STAT1 has also been detected in close proximity to E-cad in human cells using proximity biotinylation and quantitative proteomics [61], making STAT1 the likely human

STAT paralog involved. Indeed, STAT1 has an established proapoptotic EGFR-mediated function in human cells [8]. Given these high levels of evolutionary conservation of all proteins involved, we suggest that the induction of apoptosis by E-cad:EGFR:STAT complexes is likely to represent an important tumour suppressor in human epithelia.

## Materials and methods

### Key Resources table

| REAGENT or RESOURCE | SOURCE | IDENTIFIER |
| --- | --- | --- |
| **Antibodies** | | |
| Rat anti–E-cad 1:200 | Developmental Studies Hybridoma Bank | DSHB #DCAD2 |
| Rabbit anti-cleaved Dcp-1 (asp216) 1:100 | Cell Signaling Technology | Cat #9578s |
| Mouse anti-HP-1 (Su(var)205) 1:50 | Developmental Studies Hybridoma Bank | DSHB #C1A9 |
| Rabbit anti-H3K9me3 1:500 | Active Motif | Cat #39062 |
| Donkey anti-rat Alexa Fluor 488 (1:300) | Jackson ImmunoResearch | Cat #712-545-153 |
| Donkey anti-rat Alexa Fluor 647 (1:300) | Jackson ImmunoResearch | Cat #712-605-153 |
| Donkey anti-rabbit Alexa Fluor 647 (1:300) | Jackson ImmunoResearch | Cat #711-605-152 |
| Donkey anti-mouse Alexa Fluor 647 (1:300) | Jackson ImmunoResearch | Cat #715-605-151 |
| **Chemicals, peptides, and recombinant proteins** | | |
| PBS | Sigma-Aldrich | Cat #P4417-50TAB |
| Formaldehyde solution (40%) | Sigma-Aldrich | Cat #F8775 |
| Triton X-100 | Thermo Fisher | Cat #A16046 |
| BSA | New England Biolabs | Cat #B9000S |
| Vectashield mounting media | Vector Labs | Cat #H-1000 |
| Canada Balsam | Sigma-Aldrich | Cat#C1795-25ML |
| **Deposited data** | | |
| RNA sequencing data | https://www.ncbi.nlm.nih.gov/bioproject/1185032 | PRJNA1185032 |
| Microscopy data | https://qmro.qmul.ac.uk/xmlui/handle/123456789/101582 | |
| **Experimental models: Organisms/strains** | | |
| *MS1096*-Gal4;; | Bloomington Drosophila Stock Center | BDSC #8860; |
| ;;*aTub84B*-Gal80$^{TS}$ | Bloomington Drosophila Stock Center | BDSC #7017 |
| ;*patched*-Gal4; | Bloomington Drosophila Stock Center | BDSC #2017 |
| ;;*Act5C*-Gal4 | Bloomington Drosophila Stock Center | BDSC #3954 |
| ;;*UAS*-E-cad::EOS/TM6 | from ref [45] | N/A |
| ;;*UAS*-E-cad | Bloomington Drosophila Stock Center | BDSC #65589 |
| ;;*UAS*-p120catenin | from ref [35] | |
| ;*UAS*-EGFR-DN;*UAS*-EGFR-DN | Bloomington Drosophila Stock Center | BDSC #5364 |
| ;;*UAS-shg* RNAi | Vienna Drosophila Resource Center | VDRC v27082 |
| ;;*UAS-HP-1* RNAi | Bloomington Drosophila Stock Center | BDSC #33400 |
| ;;*UAS*-STAT92E$^{Y704F}$::GFP | from ref [38] | |
| ;;*UAS*-myr::GFP | Bloomington Drosophila Stock Center | BDSC #58720 |
| ;;*UAS*-CD8::Cherry | Bloomington Drosophila Stock Center | BDSC #27392 |
| ;;*UAS*-p35 | Bloomington Drosophila Stock Center | BDSC #5073 |
| ;*Egfr$^{f2}$*/ CyO; | Bloomington Drosophila Stock Center | BDSC #2768 |
| ;;*stat92E$^{397}$*/ TM3 | from ref [62] | |
| ;*wg$^{l-17}$*/ CyO; | Bloomington Drosophila Stock Center | BDSC #2980 |
| ;*arr$^2$*/ CyO; | Bloomington Drosophila Stock Center | BDSC #3087 |
| ;*hpo$^{KS240}$*/ CyO; | Bloomington Drosophila Stock Center | BDSC #25085 |
| ;;*wts$^{x1}$*/ TM6B | Bloomington Drosophila Stock Center | BDSC #44251 |

*(Continued)*

| REAGENT or RESOURCE | SOURCE | IDENTIFIER |
|---|---|---|
| *hop²*/ FM7a;; | Bloomington Drosophila Stock Center | BDSC #6032 |
| ;*POSH⁷⁴*; | Bloomington Drosophila Stock Center | BDSC #58808 |
| ;*bsk¹*/ CyO; | Bloomington Drosophila Stock Center | BDSC #3088 |
| ;;*10xStat92E*-GFP/ TM6 | Bloomington Drosophila Stock Center | BDSC #26200 |
| ;EGFR::sfGFP; | Bloomington Drosophila Stock Center | BDSC #92329 |
| ;STAT92E::GFP; | Bloomington Drosophila Stock Center | BDSC #38670 |
| ;*shg*-E-cadherin::GFP; | Bloomington Drosophila Stock Center | BDSC #60584 |
| ;*Ubi-p63E*-E-cadherin::GFP; | Kyoto Stock Center | Kyoto #109007 |
| **Software and algorithm** | | |
| Custom MATLAB scripts | https://github.com/nbul/ | N/A |
| Fiji (ImageJ) | https://fiji.sc/ | N/A |
| FV10-ASW | Olympus | N/A |
| LAS X | Leica | N/A |
| R and R Studio | https://posit.co/download/rstudio-desktop/ | N/A |
| Illustrator 20 | Adobe | N/A |
| Gimp | https://www.gimp.org/ | N/A |
| Microsoft Office 365 | Microsoft | N/A |
| **Other** | | |
| FV1000 | Olympus | N/A |
| Stellaris 8 | Leica | N/A |
| DMRA2 | Leica | N/A |
| Axioskop 2 MOT | Zeiss | N/A |

## Fly stocks and husbandry

*Drosophila melanogaster* flies were raised on standard cornmeal/agar/molasses media at 25°C unless otherwise specified. We used the UAS/Gal4/Gal80 system [43] for the overexpression and knockdown experiments, with *MS1096*-Gal4 (Bloomington 8860), *Act5C*-GAL4 *αTub84B*-Gal80$^{TS}$ (recombined Bloomington 3954 and 7017) and *patched*-Gal4 (*ptc*-Gal4) *αTub84B*-Gal80$^{TS}$ (Bloomington 2017 and 701) driver lines. The following UAS lines were used: *UAS*-E-cad::EOS [45], *UAS*-E-cad (Bloomington 65589), UAS-p120catenin [35], *UAS*-EGFR-DN (Bloomington 5364), *UAS-shg* RNAi (E-cad-RNAi, VDRC v27082), *UAS-HP1* RNAi (Bloomington 33400), *UAS*-p35 (Bloomington 5073), and *UAS*-STAT92E$^{Y704F}$ [38]. The Gal4:UAS ratio was kept constant within each experimental data set using additional copies of *UAS*-CD8::mCherry (Bloomington 27392) or *UAS*-myristoylated::GFP (myr::GFP, Bloomington 58720). The following stocks with mutant alleles were used: *Egfr*$^{f2}$/CyO (Bloomington 2768), *stat92E*$^{397}$/TM3 [30], *wg*$^{l-63}$/CyO (Bloomington 2980), *arr²*/CyO (Bloomington 3087), *hpo*$^{KS240}$/CyO (Bloomington 25085), *wts*$^{x1}$/TM6B (Bloomington 44251), *hop²*/FM7a (Bloomington 6032), *bsk¹* (Bloomington 3088), and *POSH⁷⁴* (Bloomington 58808). Finally, the following stocks with transgenes encoding for fluorescently tagged proteins were used: *shg*-E-cadherin::GFP (Bloomington 60584), *ubi*-E-cad::GFP (Kyoto Stock Center Kyoto 109007), EGFR::sfGFP (EGFR::GFP, Bloomington 92329), and STAT::GFP (Bloomington 38670). *10xStat92E*-GFP (Bloomington 26200) was used to monitor the canonical JAK/STAT activity. All experiments and relevant controls were done in parallel.

## Dissections and immunostaining

Wandering third instar larvae were dissected at 6 d after egg laying at 25°C. When *MS1096*-Gal4 was used, only male larvae were selected to ensure the same levels of Gal4 expression across all animals. Cuticles with attached imaginal

discs were fixed for 20 min with 4% formaldehyde (Sigma; F8775) in PBS (phosphate buffer saline; Sigma-Aldrich; P4417) at room temperature, washed with PBS with 0.1% Triton X-100 (Thermo Fisher; A16046; hereafter PBST), and incubated with 1% bovine serum albumin (BSA; New England Biolabs; B9000S) in PBST for 1 h at room temperature. Cuticles were incubated with primary antibodies and 1% BSA in PBST overnight at 4°C, washed in PBST, incubated for 2 hours with secondary antibodies and 1% BSA in PBST, and washed in PBST again. All antibodies and their concentrations are listed in the Key Resources table. Finally, discs were separated from the cuticles and mounted in Vectashield (Vector Labs; H-1000). Only one disc per larvae was taken at this stage. For samples not requiring immunostaining, discs were immediately mounted following fixation and one round of washing in PBST.

## Microscopy

**Confocal microscopy.** Most confocal microscopy experiments except colocalisation analyses were done using an upright Olympus FV1000 confocal microscope with either 60×/1.40 NA (fluorescence intensity and colocalisation) or 20×/0.75 NA (apoptosis) objectives. Images for STAT92E::GFP localisation, effects of p120-catenin overexpression on EGFR localisation, and cleaved Dcp-1 staining in wing discs with different doses of the *POSH* gene were obtained using an inverted Leica Stellaris 8 confocal microscope with a 63 × /1.40 NA or or 20 ×/0.75 NA (apoptosis) objectives. For fluorescence intensity and colocalisation, images were taken in the dorsal region of the wing disc pouch (Fig 1C). For imaging with x60 and x63 objectives, a single z-stack of eight slices (localisation to cell-cell borders and vesicles) or 45 slices (localisation in the nucleus) spaced by 0.38 μm was acquired for each disc capturing the complete span of the cell-cell junctions. For imaging with the x20 objective, a z-stack of 15 slices spaced by 1 μm spacing was acquired capturing the complete wing disc. All the images were at 16-bit depth in Olympus or Leica binary image format.

**Adult wing imaging.** Flies were kept overnight at -20°C, and the wings were dissected and then mounted in Canada Balsam (Sigma; C1795-25ML). Wings were imaged using a Zeiss Axioskop 2 MOT microscope or a Leica DMRA2 upright microscope with a Lumenera Infinity 5–5 camera. For both microscopes, x5 objectives were used.

## RNA sequencing

Individuals were raised at 17°C inactivating the *Act5C-Gal4* tool using temperature-sensitive tubGal80 until they reached the third instar larval stage (13 days). Larvae were then moved to 29°C for 24 hours before harvesting. RNA was extracted from wing discs dissected on ice using the NucleoSpin RNA XS Kit (Macherey-Nagel; 740902.50). Samples were dissolved in ddH$_2$O and stored at -80°C before delivery to Novogene. Input quantities were measured with Qubit Fluorometer to pre-check that they all met the company's specifications. For each of the four genotypes (*UAS*-myr::GFP, *UAS*-E-cad::EOS, *UAS*-STAT92E$^{Y704F}$::GFP and *UAS*-HP1 RNAi), three independent biological replicates were extracted. cDNA library sequencing was performed by Novogene, Cambridge, UK. The twelve samples displayed an Effective rate of approx. 99%.

## Data analysis

**Fluorescence levels at cell borders and in the cytoplasm.** Average fluorescence intensity was measured on average projections of z-stacks with native GFP fluorescence signal. To classify the signal in images into cell boundaries and the cytoplasm, we created binary masks from average projections of z-stacks with E-cad immunostaining or native fluorescence of E-cad::GFP using the Tissue Analyser plugin in Fiji [63]. These binary masks were used to measure the fluorescence intensity of average projections using our in-house MATLAB script (https://github.com/nbul/Intensity/TotalMean/). In short, each cell was identified from the mask as an individual object. The boundary of each object was dilated using a diamond-shaped morphological structural element of size 3 to encompass the XY spread of the boundary signal. The mean and total (sum of all pixel intensities) intensities of the dilated boundary (membrane signal) and the object with subtracted boundary (cytoplasm) were calculated. All the values were averaged to produce single values per

wing disc, thus testing biological replicates and excluding a variable contribution of individual discs to the result due to differences in cell numbers.

**Adult wing size.** The size of adult wings was measured by manually tracing the wing outlines using the Freehand selection tool in Fiji.

**Levels of cleaved Dcp-1.** The average intensity projections with immunostaining for cleaved Dcp-1 were analysed in Fiji. First, the signal was blurred using the Smooth filter that replaces each pixel with the average of its 3x3 neighbourhood. The background intensity was manually measured using a circular region of interest (ROI) outside apoptotic cells. The projection images were thresholded using a 2x background intensity threshold. The wing pouch area was traced in maximum projections of z-stacks with E-cad immunostaining signal using the Freehand selection tools. The selection was copied over to the threshold images of the cleaved Dcp-1 signal and analysed using the Analyze Particles Tool with the minimum size of a particle limited to 4 µm$^2$. The fraction of area covered by particles within the selection was used as the apoptosis level.

**Protein colocalisation.** To measure STAT::GFP with E-cad signal visualised by antibody, we selected Manders's colocalisation coefficients (MCC) as more informative for probes distributed to more than one compartment than Pearson's correlation coefficients [64]. The z-stacks with STAT::GFP and E-cad signal were thresholded in Fiji ([https://fiji.sc](https://fiji.sc)) by automatically setting the threshold level using Fiji's Default method and the section in the middle of the border signal visualised with E-cad antibody. The resulting two binary images were multiplied to extract the positive pixels for both STAT::GFP and E-cad. The resulting 3 binary stacks were used to calculate the percentage of STAT::GFP positive pixels that also have E-cad and the total percentage of STAT::GFP positive pixels (baseline value for random co-localisation). The difference between these two values produced MCC$_{diff}$ (%).

For colocalisation of EGFR::GFP with E-cad in the cytoplasm, we analysed the images section by section with an in-house MATLAB script ([https://github.com/nbul/Localization](https://github.com/nbul/Localization)). This script followed the same principle to obtain MCC$_{diff}$ values as above, however, the E-cad signal at cell-cell borders was excluded from the analysis using binary masks produced using the Tissue Analyser plugin in Fiji [63]. The binary masks were dilated using a diamond-shaped morphological structural element of size 3 to encompass the XY spread of the E-cad signal. The resulting MCC$_{diff}$ (%) values accounted for the intracellular signal only.

**Nuclear protein levels (individual nuclei).** The z-stacks were cropped in Fiji to contain only cells expressing CD8::mCherry, E-cad::EOS or E-cad RNAi. Average projections were made using 4 consecutive z-sections that capture the signal around nuclear centres.

For the HP1 signal, projections from images of control wing discs expressing CD8::mCherry were segmented using the cyto3 model CellPose 3.0 [65]. The segmentation errors were manually corrected, and the resulting masks were used to re-train the model. The new model was validated on images from the CD8::mCherry control wing discs and then used to segment nuclei in wing discs with E-cad overexpression and knockdown. Each image segmentation was manually verified, and errors were further corrected as needed. The generated masks were used to obtain the nuclear mean intensities on a nucleus-by-nucleus basis with an in-house MATLAB script.

For H3K9me3, the projections were analysed using an in-house MATLAB script. First, the projections were filtered using a 2-D Gaussian smoothing kernel with a standard deviation of 2. The images were binarised using an adaptive threshold that is calculated using local first-order image statistics around each pixel. This created masks with H3K9me3 puncta as individual objects. Any objects smaller than 200 px or touching image borders were removed. The above values were empirically determined from a series of tests on images of all three genotypes, and determining the most efficient combination. The masks were then used to obtain the puncta mean intensities on a punctum-by-punctum basis with an in-house MATLAB script.

The CellPose model HP1_wing and relevant MATLAB scripts are available at [https://github.com/nbul/Intensity/tree/master/HP1](https://github.com/nbul/Intensity/tree/master/HP1).

**Nuclear protein levels (average) and STAT::GFP nuclei/cytoplasm ratio.** The z-stacks were cropped in Fiji to contain only cells expressing CD8::mCherry or E-cad, and only cells not expressing CD8::mCherry or E-cad in the same wing discs. Average projections were made using 4 consecutive z-sections that capture the signal around nuclear centres. The threshold level was determined by manually measuring background intensity (outside any detectable nuclei) in the projection with HP1 signal from the area not expressing any transgene. The projections with the HP1 signal from both expressing and non-expressing areas in this wing disc were blurred using the Smooth filter that replaces each pixel with the average of its 3x3 neighbourhood. Then they were thresholded using a 2x background intensity threshold. The mean intensity within and outside the threshold area was measured for both HP1 and STAT::GFP projections.

For the HP1 signal, first, the mean intensity outside the thresholded area was subtracted from the mean intensity within the thresholded area. Then, the resulting intensity was normalised by dividing the intensity in the transgene-expressing area by that in the corresponding non-expressing area.

For STAT::GFP, first, the ratio of mean intensity within the thresholded area and outside it was produced. Then, it was normalised by dividing by the ratio in the transgene-expressing area by that in the corresponding non-expressing area.

***10xSTAT*-GFP levels.** Average fluorescence intensity was measured on average projections of z-stacks of 15 z-sections spaced by 1 μm and spanning the whole depth of the wing disc with native GFP fluorescence signal. A square 50x50 px ROI was manually positioned over overexpressing or control regions using the CD8::mCherry native fluorescence (control) or E-cad signal (E-cad overexpression) in the wing pouch area (as indicated in Fig 1C) or notum.

**RNA sequencing: Alignment to the reference genome.** RNA-seq libraries were sequenced using Illumina HiSeq. Reads were aligned to the dm6 assembly of the *D. melanogaster* genome using BWA v. 0.7.7 [66] with default settings (BWA-backtrack algorithm). The SAMtools v. 0.1.19 'view' utility was used to convert the alignments to BAM format.

**RNA sequencing: Differential expression analyses for genes.** We built an exon model based on Ensembl Gene database gene annotation. Tag counts for each gene were extracted from BAM alignment files using the HTSeq method working in union mode and implemented in R. These values were used to build an expression matrix. The differential gene expression between samples was tested using the DESeq2 package [67]. Reads per kilobase of exon model per million mapped reads (RPKM) normalised expression values were generated using the "median ratio method" (Equation 5 in [68]). We used false discovery rate (FDR)-adjusted p-values$<0.05$ and maximum posterior estimates of $\log_2$ fold change (LFC) > 1 to call genes up-regulated, and FDR-adjusted p-values$<0.05$ and LFC$<-1$ to call genes down-regulated.

**Statistical analysis and data presentation.** All statistical analyses and graphs were done in R. Venn diagrams were generated using BsRC tools (http://barc.wi.mit.edu/tools/) and re-coloured to match the colour scheme using Gimp (https://www.gimp.org/). To calculate the significance of the overlap between altered genes in RNAseq experiments and the enrichment of JNK pathway components, we used a hypergeometric distribution probability calculator (https://systems.crump.ucla.edu/hypergeometric/index.php). To compare the difference between the population mean and a hypothetical value (for colocalisation), the one-sample t-test was used. For comparing two populations, the unpaired or paired (S2 Fig only) two-sample Wilcoxon test was used for consistency as several datasets failed the Shapiro-Wilk normality test. For comparing three or more datasets (Figs 2C, S3 and S5), one-way ANOVA and post-hoc t-test with false discovery rate p-value correction were used. For testing phenotype rescue, two-way ANOVA and post-hoc t-test with false discovery rate p-value correction were used.

## Supporting information

**S1 Fig. Effects of E-cadherin overexpression on E-cadherin cell-cell border levels in the wing pouch and *10xSTAT92E*-GFP levels in the notum.** (**A**) Quantification of the E-cad mean intensity at cell-cell borders in wing discs expressing CD8::mCherry (left, black) and E-cad (right, blue). The Wilcoxon test was used to compare the datasets. N$=4$ and 8 wing discs. The representative images and other quantifications are shown in Fig 1D–1G. (**B**) Quantification of

*10xSTAT92E*-GFP levels (*10xSTAT*) in the notum of wing discs expressing CD8::mCherry and E-cad::EOS. The Wilcoxon test was used to compare the datasets. N = 4 and 4 wing discs. The representative images and the quantification in the wing pouch area are shown in Fig 1C and 1H.
(TIF)

**S2 Fig. Effects of E-cad::EOS overexpression on E-cadherin levels.** Representative images (**A**) and quantification (**B**) of cells overexpressing E-cad::EOS with *ptc*-Gal4 for 19 hours. Single confocal z-sections of the E-cad antibody staining (A, top row) and native EOS fluorescence (A, bottom row) are depicted with the depth at the middle of the adherens junctions set to 0 and decreasing as progressing from the apical to basal direction. The paired-samples Wilcoxon test was used to compare expressing (E-cad, blue in B) and non-expressing areas in same wing discs (C, black in B). N = 8 wing discs.
(TIF)

**S3 Fig. Changes in H3K9me3 localisation following E-cadherin overexpression.** Representative images (A) and quantification (B) of H3K9me3 (antibody staining) in wing discs with E-cad::EOS overexpression (OE, middle), E-cad knockdown (RNAi) or overexpression of CD8::mCherry as control (left) for 24 hours using *ptc*-Gal4. Each dot represents a single puncta. One-way ANOVA and post-hoc t-test with false discovery rate p-value correction were used in B. N = 136/3, 125/3 and 228/6 (punctum/wing disc). Similar data for HP1 is in Fig 2B and 2C.
(TIF)

**S4 Fig. Effects of E-cad overexpression on adult wing size, apoptosis and phosphorylated ERK, and no effect of EGFR-DN overexpression on E-cad levels.** (**A**) Representative images of adult wings expressing two copies of myr::GFP (top) and untagged E-cad (bottom). (**B-C**) Representative images (B) and quantification (C) of Dcp-1 staining in wing discs expressing myr::GFP (B, left) or untagged full-length E-cad (B, right column) in the presence of two functional copies (B, top row) or one copy (B, bottom row) of the *Egfr* gene. The *MS1096*-Gal4 driver was used. Two-way ANOVA and post-hoc t-test with false discovery rate p-value correction were used. N = 15, 7, 10 and 7, left-to-right. p = 0.003 for the interaction between untagged E-cad overexpression and *Egfr* copy number. Similar datasets for E-cad::EOS are in Fig 3A–3D. (**D-E**) Representative images (D) and quantification (E) of mean cell-cell border levels (E, left) and the ratio between the cytoplasm and cell-cell border levels (E, right) of E-cad::GFP expressed from a ubiquitous promoter in control wing discs (D, left, E, black) and those expressing EGFR-DN (D, right, E, red). The Wilcoxon test was used to compare the datasets. N = 6 and 6 wing discs. Similar dataset for *Egfr* heterozygotes is in Fig 3E and 3F. (**F-G**) Representative images (F) and quantification (G) of levels of phosphorylated ERK (pERK) visualised with antibody staining in control wing discs (F, left, G, black) and those overexpressing E-cad::EOS (F, right, G, blue). The Wilcoxon test was used to compare the datasets. N = 5 and 8 wing discs.
(TIF)

**S5 Fig. No rescue of the small wing phenotype by other signalling pathways than EGFR.** Representative images (**A**) and area quantification of adult wings (**B**) expressing either myr::GFP (A, top row) or E-cad::EOS (A, bottom row) in the presence of two copies (A, top) or one functional copy (A, bottom) of the following genes: *wingless* (*wg*$^{l-17}$ allele), *arrow* (*arr*$^2$ allele, encoding a co-receptor of wg), *hippo* (*hpo*$^{KS240}$ allele), *warts* (*wts*$^{x1}$ allele), and *hopscotch* (*hop*$^2$ allele, encoding for the single *Drosophila* JAK protein). The quantification is shown as a relative area to corresponding controls (one gene copy without E-cad::EOS overexpression). One-way ANOVA and post-hoc t-test with false discovery rate p-value correction were used. N = 14/12, 21/12, 11/14, 9/4 and 10/12 (one copy/two copies), left-to-right.
(TIF)

**S1 Table. Numerical data.**
(XLSX)

## Acknowledgments

We thank the Wolfson Light Microscopy and Fly Facilities at the University of Sheffield and the Microscopy Facility at the School of Biological and Behavioural Sciences, Queen Mary University of London, for their technical support.

## Author contributions

**Conceptualization:** Martin P Zeidler, Natalia A Bulgakova.

**Data curation:** Natalia A Bulgakova.

**Formal analysis:** Przemyslaw A. Stempor, Natalia A Bulgakova.

**Funding acquisition:** Natalia A Bulgakova.

**Investigation:** Miguel Ramirez Moreno, Amy Quinton, Eleanor Jacobsen, Natalia A Bulgakova.

**Methodology:** Natalia A Bulgakova.

**Project administration:** Natalia A Bulgakova.

**Supervision:** Natalia A Bulgakova.

**Visualization:** Natalia A Bulgakova.

**Writing – original draft:** Miguel Ramirez Moreno, Martin P Zeidler, Natalia A Bulgakova.

**Writing – review & editing:** Natalia A Bulgakova.

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
