## [Decision Letter · Decision Letter 0]

PGENETICS-D-24-01504

E-cadherin endocytosis promotes non-canonical EGFR:STAT signalling to induce cell death and inhibit heterochromatinisation

PLOS Genetics

Dear Dr. Bulgakova,

Thank you for submitting your manuscript to PLOS Genetics. After careful consideration, we feel that it has merit but does not fully meet PLOS Genetics's publication criteria as it currently stands. Therefore, we invite you to submit a revised version of the manuscript that addresses the points raised during the review process.

Please submit your revised manuscript within 60 days Mar 31 2025 11:59PM. If you will need more time than this to complete your revisions, please reply to this message or contact the journal office at plosgenetics@plos.org. Please include the following items when submitting your revised manuscript:

We look forward to receiving your revised manuscript.

Kind regards,

Lolitika Mandal, Ph.D

Academic Editor

PLOS Genetics

Pablo Wappner

Section Editor

PLOS Genetics

Aimée Dudley

Editor-in-Chief

PLOS Genetics

Anne Goriely

Editor-in-Chief

PLOS Genetics

**Journal Requirements:**

https://journals.plos.org/plosgenetics/s/submission-guidelines#loc-parts-of-a-submission

- ® on page: 20.

Potential Copyright Issues:

i) Please confirm (a) that you are the photographer of S5A, or (b) provide written permission from the photographer to publish the photo(s) under our CC BY 4.0 license.

**Reviewers' comments:**

Reviewer's Responses to Questions

Reviewer #1: The research uses Drosophila to address an underexplored mechanism by linking E-cadherin endocytosis to two non-canonical STAT activities - EGFR:STAT signaling in promoting apoptosis and HP1:STAT interaction in promoting heterochromatin formation. The manuscript presents a well-executed study with significant implications for understanding the biological functions and regulations of non-canonical STAT signaling pathways. The use of Drosophila as a model system ensures genetic simplicity and robust data. A combination of RNA-seq, microscopy, genetic manipulation, and immunostaining lends credibility to the conclusions. The findings are significant and should contribute to our understanding non-canonical STAT functions in developmental contexts.

I have the following minor suggestions.

Figure 1. Should provide an estimate of the levels of changes in HP1 and STAT after transgene overexpression.

Discussion: Add more discussion on the potential broader impact of the findings e.g., in tumor suppression and potential therapeutic applications.

Reviewer #2: The manuscript by Ramirez-Moreno examines potential non-canonical signaling relationships between E-cadherin, EGFR, STAT and HP1 in wing disc epithelial cells. This is a very interesting concept, and the paper is written clearly. These molecules and signaling pathways are of great interest to many, so the results could be quite impactful. That said, it is difficult to demonstrate that molecules are functioning in a non-canonical way, and much of the evidence provided here is too circumstantial or insufficient. While the results are consistent with the hypotheses, the data do not provide rigorous support for several of the conclusions. There are a number of ways that this study could be improved with more experiments, or with greater depth of analysis on the experiments they have already done.

Major concerns:

Generally, I have concerns about making conclusions about direct pathway interactions based on the genetic manipulations carried out in this paper, which are likely to have widespread effects. This pleiotropy makes it challenging to be sure whether the phenomena they describe are direct or indirect. For example, upregulating E-cadherin or mutant STAT or downregulating HP1 and waiting 24 hours may report indirect changes in gene expression and may additionally cause cell death. So some of the downstream genes may be regulated indirectly, not just by these pathways as the paper implies. It would help their case to have supporting data from other sources on the regulation of candidate genes, or additional RNAseq timepoints, or more direct evidence that the gene expression pattern changes can be seen directly in the affected cells near the time of the change the expression of the putative regulator.

A key aspect of the authors’ arguments is that E-cadherin changes its subcellular localization in response to mutant STAT overexpression and that, similarly, EGFR also changes its localization in response to E-cadherin overexpression. They use this to imply there are direct interactions between these proteins in complexes. However, I have some reservations about this, given the images and graphs shown. First of all, these proteins are generally found at the plasma membrane, but aside from that, we cannot tell if they are close enough to interact. Other methods might resolve this and would provide a more convincing argument, such as proximity ligation assays, which they seem to have most of the tools to do. The authors argue that loss at the membrane is a result of endocytosis, but this is never shown directly and raises a significant concern about the conclusions, some of which appear in the abstract and the title. While there are vesicles identified in some panels (eg, Fig 1F, 2D,E), these seem very large to be endocytic vesicles. Endocytic markers in these experiments would provide a major improvement (such as YFP-Rab proteins), or the use of mutants that block endocytosis to reverse their appearance (or directly showing images of the result attributed to the literature that p120-catenin would block this). There are graphs of quantification provided, which is helpful, but since the protein expression levels also change, which appears to be true in the images and mean intensity analysis, it would be very helpful to have an unchanging control to standardize these measurements against. Since some of the experiments are done in a stripe of overexpression, it might be informative to show and measure changes in cells on either side of this border as a point of comparison.

The authors show that HP1 levels in the nucleus changes in response to lower E-cadherin, as does the amount of H3K9me3. However, HP1 can work with many partners, so this does not necessarily mean STAT is involved. Perhaps they can look for co-localization with STAT and HP1 at a high resolution, or evaluate the genetic effects of HP1 in their assays combined with genetic changes in STAT levels.

The authors focus in the second half of the paper on genetic interactions in the wing that reduce wing size due to high levels of E-cadherin. There are a number of ways that a wing size could be modified, so while these figures support the ideas of the paper generally, more analysis/depth is needed in the characterization to make a more convincing argument. The authors show the changed size can be mediated by E-cad inducing cell death in a Egfr dependent way, and then suggest that E-cad may work through STAT and POSH for this effect. The data with STAT suppressing this phenotype are a bit weak since loss of stat on its own resulted in a larger wing, so this may be separable. Loss of HP1 also induced a small wing, but this was not tested in combination with E-cad or STAT or evaluated for effects on cell death. For POSH heterozygotes and E-cad overexpression, only size was measured, but the number of apoptotic cells would need to be counted here as well to make the argument that this pathway is mediating the effect through cell death. Given these gaps, it is hard to be convinced that there is competition for STAT in non-canonical pathways downstream of E-cadherin. If that was truly the case, overexpression with the STAT Y740F mutant and STAT wt might have differential effects on these phenotypes, which the authors could test and it could bolster their conclusions and emphasis.

To substantiate the argument that the wing effects of high E-cad are mediated by JNK signaling (and to help validate the idea that the RNAseq data is informative), the authors should look at JNK activation directly, either by assaying changes in expression of these genes/proteins or evaluating an available reporter (one option may be puc-LacZ).

Minor comments:

I suggest the authors not claim to see localization to adherens Junctions (AJ), and relabel graphs to refer to cell membrane localization

The correlation analyses are somewhat hard to understand (MCC) even after reading the methods. What is the x-axis? Is this in different genetic conditions – it appears in the panel 1D that STAT is just about everywhere, so it is a correlation if 24% of this is at the membrane (1E)? In Fig panel 2F, why is this a lower coefficient (3-7%) than in the prior figure when it appears in the figure to be more often co-localized?

Figure panels 1A and 2A seem redundant – both are probably not needed.

Reviewer #3: The study from Ramirez-Moreno et. al., describes how in epithelial cells of imaginal discs, E-cad regulates non-canonical EGFR and STAT signalling to induce cell death and enhance heterochromatin formation. The study is very interesting and sheds light on how adhesion molecules, cellular trafficking and signalling pathways (canonical or non-canonical) intersect in a dynamic fashion to coordinate physiological cellular responses and respond to changes.

The authors perform a series of genetic and RNA-seq experiments showing that increased levels of E-cad initiate a non-canonical EGFR::STAT pathway with E-cad, EGFR and STAT colocalizing in endocytic vesicles that triggers apoptosis.

There are a few minor points and suggestions the authors should consider, to strengthen their arguments in the current manuscript:

1) In most experiments presented, authors use a UAS line (e.g. UAS-myr::GFP or UAS-CD8::cherry) as control to compare to the experimental set up (e.g. UAS-E-cad::EOS). However, in Figures 3F, S4B, S4D they use a “control”, whose genotype is not specified. Since it is important to compare fly lines containing the same number of UAS lines between control vs experimental genotype (as expression levels change depending on the number of UAS lines a fly has), authors could consider providing UAS expressing controls for the above mentioned examples as well.

2) In Figures 3 and S4, authors do experiments by adding or removing one copy of the EGFR or using a dominant negative (DN) construct. It would be interesting to see the effect on imaginal discs and wing size upon overexpression of a EGFR constitutively active construct in combination with E-cad::EOS and without, as this will provide a different type of arguments on the E-cad/EGFR interaction. If the effect of EGFR-CA is too strong, authors can overexpress a wild type EGFR construct.

3) Based on their RNA seq results, authors linked JNK signalling with POSH, a scaffold protein that regulates positively TNF-JNK signalling pathway. Authors should attempt to strengthen their arguments here by testing the classical Drosophila JNK basket (via RNAi or dominant negative constructs). If experimentally easy, authors can also compare the levels of puckered (via established TRE or puc-GFP lines) before and after rescue in wing imaginal discs.

4) The authors link the activation of apoptosis upon E-cad overexpression (through the non-canonical EGFR::STAT pathway) to a tumour-suppressive, cancer-preventing mechanism. This is a valid hypothesis and interpretation for the discussion as this holds true in several tissue contexts, also in relation to E-cad. Yet the way it is currently presented, as a mechanism in their own experimental conditions, in the abstract (“we propose that E-cadherin endocytosis controls the balance between two non-canonical STAT activities in a potential tumour-suppressive mechanism”) and the discussion (“We propose that the apoptosis induced by E-cad endocytosis and EGFR:STAT92E activation provides a mechanism for the elimination of potential cancer cells undergoing cell-cell adhesion disassembly during epithelial-to-mesenchymal transitions and...”), it sounds like an “overstatement” not supported by experimental evidence.

Authors can modify the text to clearly separate experimental evidence from hypothesis-driven conclusions or provide experimental evidence to support this e.g. by knocking down or removing gene copies of pro-apoptotic or apoptotic genes and see whether this leads to tumour formation, overgrowths or benign over-proliferation that disrupts wing development.

Minor comments:

1) Figures 4, 5, and Supplementary Figures are missing descriptive titles that would facilitate the readers.

2) Figure 2F is missing an axis description, in Figures 3H and S4E please add the genotypes as well, as it help readers grasp the experiment quicker.

**Have all data underlying the figures and results presented in the manuscript been provided?**

Reviewer #1: Yes

Reviewer #2: Yes

Reviewer #3: Yes

PLOS authors have the option to publish the peer review history of their article (what does this mean? ). If published, this will include your full peer review and any attached files.

**Do you want your identity to be public for this peer review?** For information about this choice, including consent withdrawal, please see our Privacy Policy .

Reviewer #1: No

Reviewer #2: No

Reviewer #3: **Yes: ** Dr Fani Papagiannouli

**Figure resubmission:**
---

## [Decision Letter · Decision Letter 1]

Dear Dr Bulgakova,

We are pleased to inform you that your manuscript entitled "E-cadherin endocytosis promotes non-canonical EGFR:STAT signalling to induce cell death and inhibit heterochromatinisation" has been editorially accepted for publication in PLOS Genetics. Congratulations!

Yours sincerely,

Pablo Wappner

Section Editor

PLOS Genetics

Pablo Wappner

Section Editor

PLOS Genetics

Aimée Dudley

Editor-in-Chief

PLOS Genetics

Anne Goriely

Editor-in-Chief

PLOS Genetics

Comments from the reviewers (if applicable):

Reviewer's Responses to Questions

**Comments to the Authors:**

Reviewer #1: I appreciate the authors' effort in addressing the issues raised, and I'm satisfied with the revised manuscript.

Reviewer #2: The revised manuscript by Ramirez-Moreno is improved and includes additional experiments to substantiate their claims that STAT participates in non-canonical signaling with E-cadherin and EGFR or HP1 in wing disc epithelial cells. Most of my concerns were resolved by new data, analysis, or further explanations, and so the work is quite rigorous. Since these are very common pathways that have not been shown to be in this relationship before, and there is a likely connection to tumor suppression mechanism, the paper will be of broad interest.

The authors do a nice job testing how the two non-canonical pathways balance each other out. I had a minor suggestion. Since most of the experiments (with a few exceptions) use cells that do not have the canonical signaling, the authors might consider emphasizing that in cells that do not have the canonical signaling, STAT participates in other required functions, which is certainly interesting in its own right, instead of focusing on the canonical signaling being dominant. On a related note, I suggest they re-consider how figure 9 represents their case - currently the schematic gives the impression of equilibrium or equal weight to each direction of the signaling.

A few other minor comments:

When E-cad is overexpressed, are the cells bigger? (eg, panel 1D, panel 3E). if so, can the authors mention this in the text?

Does fig 2A use ptc-Gal for the overexpression?

The legend for figure 4 has B/C swapped.

Reviewer #3: I would like to thank the authors for addressing reviewer comments and therefore I suggest the acceptance of the manuscript at this stage without any further changes.

The study is very interesting and sheds light on how adhesion molecules, cellular trafficking and signalling pathways (canonical or non-canonical) intersect in a dynamic fashion to coordinate cellular responses and respond to changes.

**Have all data underlying the figures and results presented in the manuscript been provided?**

Reviewer #1: Yes

Reviewer #2: Yes

Reviewer #3: Yes

PLOS authors have the option to publish the peer review history of their article (what does this mean? ). If published, this will include your full peer review and any attached files.

**Do you want your identity to be public for this peer review?** For information about this choice, including consent withdrawal, please see our Privacy Policy .

Reviewer #1: No

Reviewer #2: No

Reviewer #3: No

**Data Deposition**

http://datadryad.org/submit?journalID=pgenetics&manu=PGENETICS-D-24-01504R1

**Press Queries**

---

## [Editor Report · Acceptance letter]

PGENETICS-D-24-01504R1

E-cadherin endocytosis promotes non-canonical EGFR:STAT signalling to induce cell death and inhibit heterochromatinisation

Dear Dr Bulgakova,

We are pleased to inform you that your manuscript entitled "E-cadherin endocytosis promotes non-canonical EGFR:STAT signalling to induce cell death and inhibit heterochromatinisation" has been formally accepted for publication in PLOS Genetics! Your manuscript is now with our production department and you will be notified of the publication date in due course.

With kind regards,

Zsuzsanna Gémesi

PLOS Genetics

On behalf of:
